



# Improvement of the thermal spring protection area through numerical modelling and interdisciplinary studies.

Joaquín Sanz de Ojeda[1,*], Francisco Javier Elorza Tenreiro [2] and Eugenio Sanz Pérez[3]

[1] Escuela Técnica Superior de Ingenieros de Minas y Energía, Universidad Politécnica de Madrid, c de Ríos Rosas, 21, 28003,
   Madrid, Spain; joaquin.sanzdeojed@alumnos.upm.es

[2] Escuela Técnica Superior de Ingenieros de Minas y Energía, Universidad Politécnica de Madrid, c de Ríos Rosas, 21, 28003,
   Madrid, Spain; franciscojavier.elorza@upm.es

[3] Laboratorio de Geología, Departamento de Ingeniería y Morfología del Terreno, Universidad Politécnica de Madrid, c Profesor
   Aranguren s/n, 28040, Madrid, Spain; eugenio.sanz@upm.es

*  Correspondence: joaquin.sanzdeojed@alumnos.upm.es

**Abstract.** The integration of different sources of geological and hydrogeological information and the application of interdisciplinary methods have informed the origin of the thermal springs of Alhama de Aragón and Jaraba, as well as others associated semi thermal springs (1,200 l/s of combined flow, 711 l/s at over 30ºC).

This issue is key to being able to design any sustainable conservation strategy in terms of quantity and quality of resources within the recharge area of the most important thermal springs in Spain.

The Upper Cretaceous limestones and dolomites constitute the main aquifer of the Alhama and Jaraba thermal system. It extends continuously under the slightly permeable Tertiary of the Almazán Basin in the form of a NW-SE "synclinorium". Its bottom has extensive depths of more than 3,000 and 4,000 m in the NE sector, which constitute the focus of heat considering normal geothermal gradients.

From the results of the modelling of the flow of this thermal system, it can be concluded that the origin of these springs comes mainly from the autogenous recharge that occurs in the Cretaceous calcareous outcrops that border the Almazán Basin to the north, both in the Ebro Basin (Jalón Valley) and in the more distant Duero Basin. The underground flow follows a NW-SE direction flowing across the Duero-Ebro divide, favored by the topographic difference in elevation between the two basins. The modelled regional flow is coherent with the progressive increase in temperature, water age, mineralization, and flow of the springs through which the system discharges.

**Keywords**. Delimitation of protection zone. Conservation of thermal springs. Numerical flow modelling. Tritium isotopes.

## 1 Introduction and objectives

The sustainable development of groundwater resources and their management largely depends on the knowledge of aquifer systems, spatial and temporal groundwater recharge and discharge rate, as well as groundwater storage. Demographic changes and population growth in various regions and their corresponding increase in demand for groundwater to meet drinking water needs, as well as the impact of climate change on groundwater conditions are other factors that have to be studied. to ensure sustainable management of groundwater. Some thermal aquifers, like the one studied here, are aquifers of special value,





since they support the concentration of the most important and numerous spa establishments in Spain. But its origin was still unknown. It is therefore very interesting to know the feeding area of these springs in order to manage preventive conservation hydrogeology in them to avoid problems of unsustainable exploitation or contamination in the future, as has happened in other or be prepared for climate.

On the other hand, and as is known, the protection perimeters around a water collection for urban consumption are intended to protect the quality and quantity of water from the potential risks of contamination and subtraction of flows, which can represent anthropogenic activities in the vicinity of the catchment. The most common protection system consists of dividing the catchment environment into different zones, graded from highest to lowest risk and importance with respect to the restriction of other activities.

Regarding the protection perimeters, in detrital and mixed aquifers the methodology used is usually based on the use of analytical methods among which the Wyssling method predominates (Wyssling, 1979 in Lallemand-Barrés and Roux, 1989), designed for the sizing of protection perimeters in aquifers with intergranular porosity and homogeneous. On the other hand, in karst aquifers, given the uniqueness and specificity of their hydrogeological functioning, the analytical methods used in non-karstified environments for the estimation and definition of protection perimeters do not offer results that guarantee their functionality and effectiveness. This is due to a series of specific peculiarities that differentiate them from the rest of the aquifer typologies (Goldscheider et al., 2004). To this we can add the complication of knowing the origin of the thermal springs associated with these calcareous lithologies, since they usually come from deep areas that are difficult to investigate.

In this context, personalized or particularized hydrogeological analysis for each specific case takes special relevance, where it is very important to know the conceptual model of hydrogeological functioning. In the case studied, it was essential to know the origin and recharge area first as a first step before proposing a practical and concrete protection perimeter. That is why an interdisciplinary study has been carried out in which all existing data from different sources have been used to develop a conceptual model and then a numerical simulation. All of this is particularly applicable to urban supplies, underground table water bottling plants and spas. All these uses occur in the case studied, especially because it constitutes the largest concentration of spas in Spain.

The springs of Alhama de Aragón and Jaraba (Zaragoza) along with other upwellings associated with the same geological area, form the low-temperature hydrothermal system with the largest groundwater resources in the Iberian Peninsula and one of the largest in Europe. (Fig. 1) (Sanz y Yélamos 1998). The total flow are around 1,200 l/s ($37.8 \text{ hm}^3$/year), distributed between the springs of Jaraba (600 l/s, although 322 l/s is thermal flow, the rest is cold water, altitude 760m), Alhama de Aragón (434 l/s, altitude 660m), Deza (140 l/s, altitude 910 m), San Roquillo (10 l/s, altitude 900 m), and Embid de Ariza (30 l/s, altitude 800 m) (ITGE-DGA, 1994; Sanz and Yélamos, 1998). The water temperature varies between 18º and 34ºC: in Deza the water rises between 18º and 20ºC, in Embid de Ariza at 29ºC, in Alhama de Aragón between 27º and 34ºC, and in Jaraba between 18º and 32ºC. The flow above 30ºC is the largest and is estimated to be around 711 l/s, which represents 60% of the total flow in the system. This last flow represents 38 % of all the thermal springs over 30ºC in the Iberian Peninsula, according to the data provided by Yélamos and Sanz Pérez (1998). There are



also other springs in the Almazán Basin, which is part of the Duero basin, with temperatures between 4ºC and 7ºC higher than the average temperature of the springs in the area, which is around 11ºC (Fig 1).

The springs of the thermal system of Alhama de Aragón and Jaraba were preferred in ancient times as supplies due to their high guarantee and the quality of their waters. In a region as arid as this one, where the average

rainfall barely exceeds 450 mm per year, the low depletion coefficients of the springs meant that their flows remained quite constant and with a lot of inertia, which offered insurance of supply during the seasons. Thus, for example, the Deza springs were used to supply water to the city and military camps of Titiakos (ancient Deza) in Celtiberian and Roman times (Sanz Pérez et al., 2023). The thermal and mineral-medicinal waters were used for therapeutic purposes since Roman times (Gonzalo Ruiz, 2001 and Lagüens Cobo, 2002).

There are currently seven renowned spas and three natural mineral water bottling plants, making it up one of the most important balneotherapy complexes in Spain. In Alhama de Aragón, these water settlements use about 300 l/s, the rest being discharged directly into the river Jalón, which flows through the vicinity. In Jaraba, there is a total flow of 600 l/s which arises through diffuse discharge and springs along a stretch of the river of 2km from the river Mesa. From reports by ITGE-DGA (1994) it can be deduced that this flow is

distributed between 322 l/s with a temperature of 31-32ºC, and 277 l/s is cold water that comes out at 12ºC. The hot springs of Jaraba are used by spas and bottling plants.

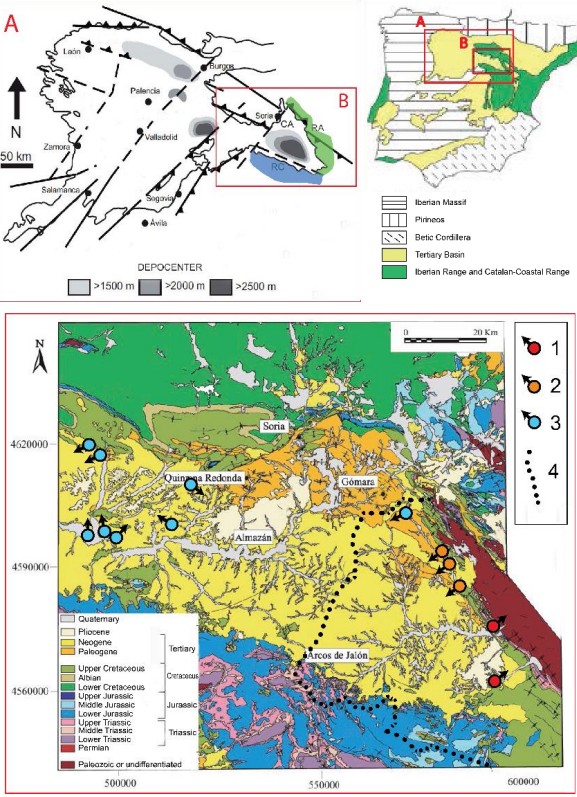

**Fig. 1.** *Location map of the study area. A. Areas of maximum Tertiary sediment accumulation in the Duero Basin (according to Alonso Gavilán et al., 2004) and location of the Almazán Basin (CA),*





*Aragonese branch (RA) and Castilian branch (RC) of the Iberian Range. B. Geological diagram of the Almazán basin (Modified from Rey Moral (2003) and Huerta, 2007), and location of the main groups of springs: (1) thermal, over 30ºC, (2) semi-thermal, between 20 and 30ºC and (3) those with temperatures between 4ºC and 8ºC above the average temperature of the springs in the area, which is about 11ºC (4) Water divide between the Duero-Ebro and Ebro-Tajo Basins.*


The thermal and semi-thermal springs of this system are distributed along an alignment of Upper Cretaceous limestone in a NW-SE direction over a length of 57 km and with a maximum difference in height of 350 m between the highest and lowest source. This limestone band is on the edge of the Aragonese Branch of the Iberian Range in contact with the Tertiary Almazán Basin. Sixteen kilometers further northwest of Deza there is another spring (Almazul, 10 l/s, 1,000 m) which has a temperature of 16ºC and can also be considered as

the first semi-thermal manifestation. To the SE, at the opposite end, are the springs of Alhama de Aragón (660 m) and Jaraba (750 m), the latter located at the end of the synclinorium which structures the Cretaceous of the Almazán Basin and which outcrops in an arc as a periclinal closure in the Ibdes-Jaraba area (Fig. 1 and 3). The Almazán Basin has a synclinal shape with a depocenter, near the village of Almazán, in which the Tertiary sediments reach a thickness of 4500 m, thus representing one of the strongest non-marine Tertiary

series in the Duero Basin and in the Iberian Peninsula (Fig. 1).

The existence of a geothermal system requires, in addition to a heat source, the presence of permeable geological units that form aquifers or reservoirs, as well as adequate water recharge to the system. In this sense, it is essential to know and study the hydrogeological aspects in these geothermal systems (e.g., Cappacionni, et al., 2011, Chandrajit et al., 2013 for isotopic and other techniques). Or in Szocsa et al., 2018,

where a multidisciplinary study is also carried out to investigate one of the most important thermal aquifers in Europe. Regarding the background of the aquifer that is being studied, it refers to the preliminary work of Hernández Pacheco (1954), followed by SGOP (1990) and ITGE-DGA (1994). However, despite these works, the origin of the springs and the conceptual hydrogeological model of the Alhama-Jaraba thermal system are still not clearly established. Initially it was thought that the main flow of thermal water came from

the south (Sierra del Solorio), where there is a plateau with extensive outcrops of mainly Jurassic carbonate materials (Hernández Pacheco, 1954, ITGE-DGA (1994), Tena et al. (1995) and Sánchez et al., 2000). In other words, according to these authors, the water would be recharged in the Castilian branch of the Iberian Range and would come out through the aforementioned springs of different altitudes distributed along the Cretaceous alignment on the other edge of the Almazán basin (Fig. 2). Some of these authors leave open the

possibility that part of the Alhama water may also come from the northwest of the Deza (Hernández Pacheco (1954; Tena et al. (1995). It is SGOP (1990) and Sanz and Yélamos (1994) who propose that the main recharge area of Alhama is distributed throughout the Cretaceous calcareous remnants and poljes of the entire northwestern Deza and explain this in a simplified 2D model. Finally, Coloma et al. (1999) propose that recharge occurs throughout the Tertiary Almazán basin, flowing into the Cretaceous limestones at the base of

the basin and discharging into the Cretaceous outcrops at the margins.

It is understood that this uncertainty and disagreement was partly due to the lack of hydrogeological information available, in particular the lack of knowledge of the Cretaceous and its geometry at depth beneath the Tertiary of the Almazán Basin. There was also a lack of data on piezometry and deep boreholes in the Cretaceous. This lack of information has been compounded by the fact that the area is sparsely populated,

and groundwater use is low. However, since the last general hydrogeological works by Sanz and Yelamos





(1998) and ITGE-DGA (1994) 25 years ago, the authors of the present study have been able to generate and use new information, as well as to make use of existing information, all in an attempt to improve knowledge of the aquifers that give rise to these springs. As a result of this uncertainty about the origin and hydrogeological functioning of the springs of Alhama de Aragón and Jaraba and other springs associated with the same geological area, a mathematical model has been developed to understand and justify the underground flow of this geothermal system.

Thus, the geographical area of study is delimited by the hydrogeological areas that were considered a priori to be part of the thermal system of the Alhama and Jaraba springs, or that could interact with it. In this way, and from a geological point of view, the study area covers the entire Tertiary and Cretaceous of the Almazán Basin, between the hydrogeological divide between the Ebro and Duero basins, up to the peripheral Cretaceous edges of the Aragonese branch of the Iberian Range. It also includes the Jurassic-Cretaceous aquifers of the Castilian branch. This hydrogeological divide does not coincide with the Duero-Ebro hydrographic divide for the Cretaceous limestone hydrothermal aquifer, but goes far beyond it, entering the Duero basin. It should be borne in mind that it is the same aquifer, and it has been important to distinguish the separation of flows for both hydrogeological basins based on existing knowledge. For this reason, we have also extended the scope of the study to part of the Rituerto and Morón basins (rivers belonging to the Duero basin), which follow the Cretaceous in an apparent hydraulic connection beyond the hydrographic divide with the Duero. In addition to these areas, it also includes the Cretaceous and Jurassic aquifers on the edges, i.e., the Sierras de la Pica, Castejón, Miñana and Cetina in the Aragonese Branch. And Sierra del Solorio, Sierra Ministra, Tierra de Medinaceli and Altos de Radona, in the Castilian branch. This represents a vast territory with a surface area of approximately 3,500 km$^2$ in the centre-northeast of the Iberian Peninsula. This area includes the headwaters of the Jalón river (Ebro River Basin) and the dividing areas with the Duero River to the west and the Tajo River to the south (Fig. 1).

The main objective of this study is to improve the knowledge of the origin of the Alhama hot springs and their hydrogeological functioning. The integration of different sources of information, including conventional geological and hydrogeological methods, as well as environmental isotopes and simulation techniques, aims to characterize the groundwater dynamics and numerically model the flow of this geothermal system. A conceptual model of hydrogeological functioning will first be defined. Subsequently, the geothermal system has been reproduced for the first time by numerical modelling of the groundwater flow in 3D using hydrogeological modelling software. The main objectives of the modelling are as follows:

- Create a mathematical model of the area that gives validity to the conceptual model explained, in order to give numerical support to all the conceptual justifications. From this numerical model, the calibrated hydrogeological and geometric parameters will be obtained so that they are consistent with the conceptual model.

- Orienting the conceptual model from the calibration process of the numerical model, thereby providing answers or new insights into the hydrogeological functioning of the system, which would not have been possible without the mathematical model.



## 2 Description of the geology and hydrogeology of the study area

### 2.1 Geographical location, hydrography and climate.

The study area is a high area at an altitude of about 1,000 m and forms the dividing line between the Duero and Ebro basins (upper basin of the river Jalón). To the north of the Almazán Basin, the mountain ranges are oriented in a NW-SE direction, forming the edge of the Aragonese branch of the Iberian Range (1,313 m as the maximum relative altitude in the Sierra de Miñana). To the south there is a high plateau that forms the Castilian branch of the Iberian Mountain range.

From the hydrographic point of view, most of the study area belongs to the upper basin of the river Jalón, which divides the basin of the Duero to the west and the basin of the Tagus to the south.

The dividing line between the basin of the Duero and the Ebro (Alto Jalón) is located in the high plateaux of the Almazán Basin at around 1,100-1,150 m and in the reliefs that protrude from it at an altitude slightly over 1,300 m. The Jalón and its left margin are rivers with an enormous erosive capacity, in such a way that they

have captured more than 3,000 km$^2$ from the high plateau of the Duero basin by the retreat of their headwaters during the Quaternary (Sáenz Ridruejo, 1985). This is not the case with the tributaries on the right bank of the river Jalón (rivers Mesa 54 Km, 27 m$^3$/s and Piedra, 76 Km, 24 m$^3$/s for example), which are lower energy and quite regulated by large karstic springs.

The average rainfall of the Jalón river basin for the period 1920-2002 was 437 mm/year, varying between

350 mm/year in the area closest to its mouth and 555 mm/year in the areas closest to the Sierra Ministra-Sierra del Solorio at its headwaters (CHE, 2007). The average air temperature varies between 7ºC in the highest areas, and 10ºC in the sub-basins of the Mesa and Piedra rivers.

### 2.2 Geology

From a geological point of view, the study area covers the entire Tertiary of the Almazán Basin between the

hydrographic (and largely hydrogeological) divide of the Ebro with the Duero Basin up to the peripheral Cretaceous edges of the Aragonese branch of the Iberian Range, and the Jurassic-Cretaceous of the Castilian branch. The scope of the study has also extended to a part of the Rituerto Basin following the Cretaceous in apparent hydraulic connection beyond the hydrographic divide with the Duero river. In addition to these areas, it includes the Cretaceous and Jurassic aquifers on the edges, i.e., the Sierras de la Pica, Castejón,

Miñana and Cetina in the Aragonese branch. And Sierra del Solorio, Sierra Ministra, Tierra de Medinaceli and Altos de Radona, in Castilian branch.



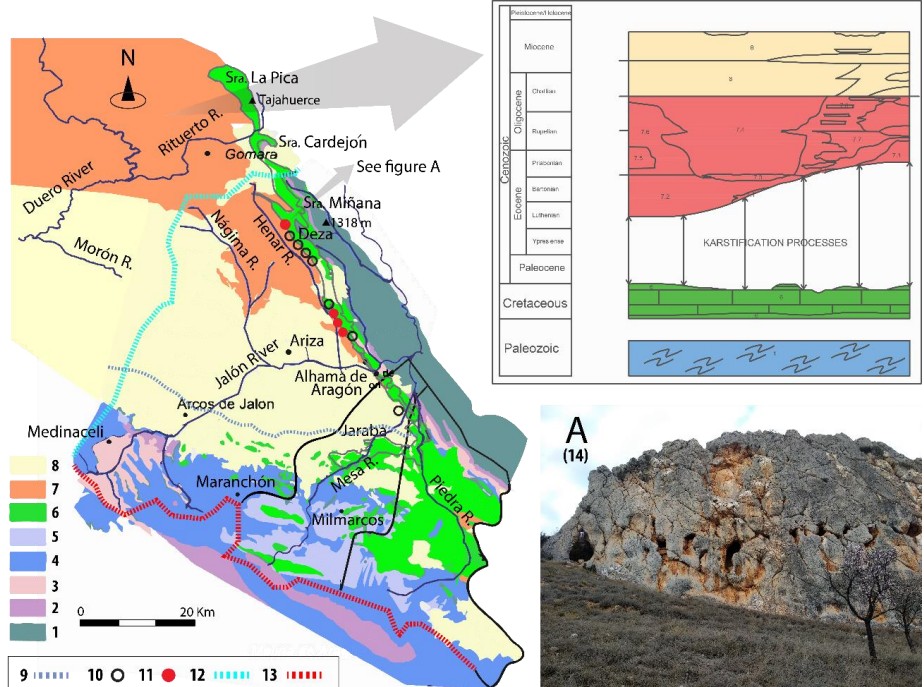

**Fig. 2.** *Geological scheme of the study area with indication of the Jurassic extension below the Cenozoic, detail of the Palaeogene stratigraphy and location of the karstification processes in the Cretaceous.*

*Geology: 1. Precambrian and Paleozoic: quartzites and shales. 2. Lower and Middle Triassic: sandstones of the Buntsanstein facies and dolomites and marls of the Muchelcalk facies. 3. Upper Triassic: clays and gypsum of the Keuper facies. 4. Lower Jurassic: Dolomites. 5. Middle and Upper Jurassic: limestones and marls. 6. Cretaceous: Utrillas facies sands below. Upper limestones and marls. 7. Palaeogene of the Northern Zone (adapted from Huerta 2007). (7.1. Ocino Fm., 7.2. Almazul Fm., 7.3. Gomara Fm., 7.4.*
*Gomara Fm., 7.5. Peroniel Fm., 7.6 Fm., 7.7 Bordalba Fm., 7.8 Deza Fm., 7.9 Valdehurtado Fm., 7.10 El Raido, 7.11 Alparrache Fm.). 8. Neogene: shales, siltstones, conglomerates. 9. Jurassic boundary below the Cenozoic. 10. Traces of pre-Palaeogene karstification at the Cretaceous-Tertiary contact (according to Armenteros, 1989). 11. Traces of karstification due to the circulation of thermal and semi-thermal waters at the Cretaceous-Tertiary contact (according to Hernández Pacheco, 1954 and this study). 12. Water*
*divide between the Duero-Ebro. 13. Water divide between the Ebro-Tajo. 14. Overview of signs of karstification at the Cretaceous-Tertiary contact.*

### 2.2.1 Stratigraphy

In the study area, the Triassic materials outcrop in their three characteristic facies of the Iberian Range (Buntsandstein facies: conglomerates and sandstones, with a thickness greater than 300 m. Muschelkalk
facies, dolomites, totaling a thickness of about 100 m. Keuper facies: 200 m of clays, silts and gypsum). Subsequently, the marine carbonate Jurassic, the Upper Cretaceous, from the Utrillas sand facies to the marine and continental carbonate series, and the Tertiary that fills the Almazán Basin. For reasons of hydrogeological interest, we will focus on a more detailed description of those permeable formations most closely related a priori to the Jaraba and Alhama de Aragón thermal springs aquifer. That is to say, the Jurassic and Cretaceous
limestone mainly.



##### 2.2.1.1 *The Jurassic in the Castilian branch of the Iberian Range. The absence of Jurassic at the edge of the Aragonese branch and under the Tertiary of the Almazán Basin.*

The Jurassic is made up of carbonate formations of marine origin outcrop and extend in the Castilian branch to the south of the study area, at the headwaters of the Jalón river. These reach a total thickness of between 310 and 335 m

The Jurassic is quite permeable and forms part of the recharge area of the Sierra del Solorio in the Aragonese branch of the Iberian Range, which is the area in which the Jaraba and Alhama springs were thought to be fed. It is anticipated, however, that within the study area the Jurassic is absent in most of the Aragonese branch of the Iberian Range and below the Tertiary of the Almazán Basin within the study area, so it will be left out of the hydrogeological game and its importance in this sense will lose interest.

Figure 2 shows the approximate limit of the Jurassic distribution below the Tertiary in the region. In all this area where there is no Jurassic, the Cretaceous (Utrillas and Santa María de las Hoyas Facies) directly overlies the Triassic. This absence of Jurassic in most of the study area will be of great importance in the definition of the hydrogeological conceptual model.

##### 2.2.1.2 *Cretaceous*

In discordance over all the aforementioned terrains are the sands and sandstones of the Utrillas facies, as well as the sands, clays and limestones of Santa María de las Hoyas. These formations have a combined thickness of about 200 m (2 in Fig. 2).

The Cretaceous carbonate formations constitute the main thermal aquifer, and their thickness generally exceeds 300 m, although it varies from area to area. According to geological sheet data, and although the entire stratigraphic series does not always outcrop, the thickness varies between 545 m in the Soria area, 1,178 m in the Sierra de la La Pica and Sierra de Tajahuerce (Esnaola and Martin Fernández, 1973; Huerta, 2007), 415 m in the Deza area, 390 m in Alhama de Aragón and 340 m in Jaraba. This carbonate Cretaceous is represented by 3 in Fig 2.

The boundary between Cretaceous and Palaeogene rocks differs from area to area. In most cases, it is an unconformity on which the Palaeogene rocks rest. In other cases, it is a paraconformity characterized by caliche cover and in other cases a surface with karstifications (Armenteros, 1989).

##### 2.2.1.3 *The Tertiary in the eastern sector of the Almazán Basin*

The stratigraphic scheme in Fig. 2 shows the distribution of the Palaeogene and Miocene mega sequences of the north-eastern sector of the Tertiary of the Almazán Basin (Huerta, 2007), where it can be appreciated their geometrical distribution and interrelation with other facies, the lateral changes, and which ones are directly supported by the Cretaceous aquifer.

#### 2.3 *Hydrogeology.*

In accordance with the stratigraphic succession of the area, the presence of two hydro stratigraphic formations is recognized to behave as important aquifers: one is formed by carbonate rocks from the Jurassic, and that constitutes the karstic system of the Sierra del Solorio within the Alto Jalón water body (CHE, 2007). And





another formed by carbonate rocks from the Upper Cretaceous which form the hydrothermal aquifer. The Tertiary of the Almazán Basin forms a powerful series with little permeability, although some aquitard sections are locally differentiated within the Palaeogene (Fig. 3).

### 2.3.1    *Hydrogeology of Sierra del Solorio*


Sierra del Solorio is a geographical region comprising the westernmost karstified Jurassic limestone and dolomite moors of the Alto Jalón, which act as a watershed between the Ebro, Tagus, and Duero basins, and the borders of the provinces of Guadalajara, Soria and Zaragoza. It includes the headwaters of the river Jalón and its main tributary, the river Mesa. Before the confluence with this river, the Jalón is joined by the river

Blanco and other less important streams, such as the Sagides, Chaorna and Iruecha (Fig. 3). Due to its relative proximity to the springs of Jaraba and Alhama, the Sierra del Solorio is the area where it was first thought that the water from these thermal springs could come from (Hernández-Pacheco, 1954). This is why we are going to describe the main hydrogeological characteristics of this unit, although there is still a lack of information, as the area is unpopulated, there is hardly any demand for water and therefore there are few

wells.

From the hydrogeological point of view, and officially, the Sierra del Solorio forms part of a larger groundwater body (09.86 - Páramos del Alto Jalón), which also includes the headwaters of the rivers Piedra and Martin (ITGE, 1980, 1981 and 1987; CHE, 2023). This groundwater body contains a wide extension of Mesozoic carbonate outcrops which are mainly included in the Castilian branch of the Iberian Range, and

which in the Sierra del Solorio are almost exclusively from the Lower Jurassic.

The structure of the Sierra del Solorio karst is a gently folded tabular peneplain karst, the flow of which is conditioned by the impermeable base of the Keuper facies and the topography. Due to the manner and type of its water supply, it can be classified as an autochthonous karst, whose pluvial and pluvio-nival supply is essentially carried out on the absorption surface of the limestone moorlands. The feeding area of this aquifer,

up to the river Mesa, is about 576 km$^2$ (SGOP, 1990) (De Toledo and Arqued, 1999). The Jurassic soils, especially the dolomites and limestones of the Lías, form the main aquifers. The strength of these varies depending on whether other upper Jurassic levels are represented or not. Most frequently they are about 200 m thick, although it increases toward the river Mesa, and decrease toward Medinaceli, where the impermeable Keuper already outcrops in the valleys, hydraulically separating some areas from others. The Cretaceous

limestone, except for the 27 km$^2$ of outcrops south of Jaraba where the Mesa river crosses them and forms a canyon, is barely represented in small islands on the northern edge of the Jurassic or emerging from the Tertiary, with a surface area of barely 4 km$^2$, i.e., the Cretaceous calcareous outcrop on the edge of the Castilian branch of the Iberian Range is 5% of the total calcareous surface area, the remaining 95% corresponds to the Jurassic.

The compartmentalization of this peneplain by the drainage network of the tributaries of the Jalón means that it does not behave as a large karst system, but rather as more or less isolated subsystems. Thus, and according to (SGOP, 1990), this hydrogeological unit can be subdivided into two sectors: an eastern sector which forms the more continuous plateau of the Paramos of Maranchón, and that drains to the great spring of Mochales (more than 500 l/s (SGOP, 1990). And a western sector coinciding with the Medinaceli area, which is more



fragmented in aquifers separated from each other. More than ten of these calcareous outcrops can be counted here, isolated from each other by the Keuper, which occupies the lower areas and valleys of the drainage network. Each of these small aquifers is associated with one or more springs, which represent its drainage.

According to the CEH (2023), this sector of the body borders by closed contact (zero flow) with other groundwater bodies located to the west and south, which belong to the Tagus and Duero basins. To the north

are the Tertiary sediments of the Almazán Basin, which according to CEH are not considered groundwater bodies (CEH, 2023).

Sierra del Solorio operate almost naturally, as there are hardly any areas of exploitation (less than 2 hm3/year in catchments for supplying small villages). Recharge is mainly through infiltration of rainwater in the outcrops of the moorland, although there are also important permanent drains in the course of the Mesa River,

where some 200 l/s seep after the Mochales springs and before reaching Jaraba. Jaraba's cold flow is largely due to these losses from the river Mesa, although there is also natural recharge from the 27 km$^2$ of Cretaceous limestone outcrops in the surrounding area. There are not enough wells in the area, so it is not possible to draw a general groundwater contour map, but the flow is directed in each sector toward various springs with sometimes quite significant flows. Referring to the flows of these springs in the low water year 1988-89

(SGOP, 1990) for which data is available, we can cite in the eastern sector the Mochales spring on the river Mesa, located at an altitude of 980 m (696 l/s). In the area of Medinaceli, the sources of the Jalón and Fuencaliente streams in Ambrona (140 l/s in total), Urex (120 l/s) and Layna (60 l/s) on the river Blanco, mainly stand out. There are other less important ones which feed the streams of Chaorna (16 l/s), Iruecha (6 l/s) and Sagides (26-37 l/s).

For an average year, the resources of the Sierra del Solorio are evaluated with the sum of all the above-mentioned springs and others of lower rank, as well as discharges to rivers, and are estimated at some 63 hm$^3$/year (CHE, 2023), of which 47 hm$^3$/year correspond to the springs of the Mesa River and some 16 hm$^3$/year to the area of Medinaceli (SGOP, 1990).

### 2.3.2 The Upper Cretaceous thermal limestone aquifer

The Cretaceous carbonate formations constitute the main thermal aquifer, and their thickness generally exceeds 300 m, although it varies from area to area. In this area of the Iberian Range around the Tertiary Basin of Almazán in the Duero Basin, they are one of the most important hydrostratigraphic formations and form very important karst aquifers that drain very abundant springs, such as those of the Fuentona de Muriel (Pérez Santos, 2007) or the Grandes de Gormaz springs (Távara Espinoza, 2011), for example.

The karstification of these shallow peripheral systems has its origin in dissolution processes that occurred mainly during the recent Tertiary and Quaternary (Rodríguez García, 2008). However, it is necessary to know whether the thermal system studied here, which reaches a depth of 4,500 m, has also been affected by these or other older dissolution processes, since the degree of porosity that these calcareous formations may reach makes it a true aquifer. On the other hand, it is also essential to know the geometry of the Upper Cretaceous

at depth and its continuity and degree of hydraulic connection, as this can have an important influence on the subterranean flow within the same hydrostratigraphic formation.





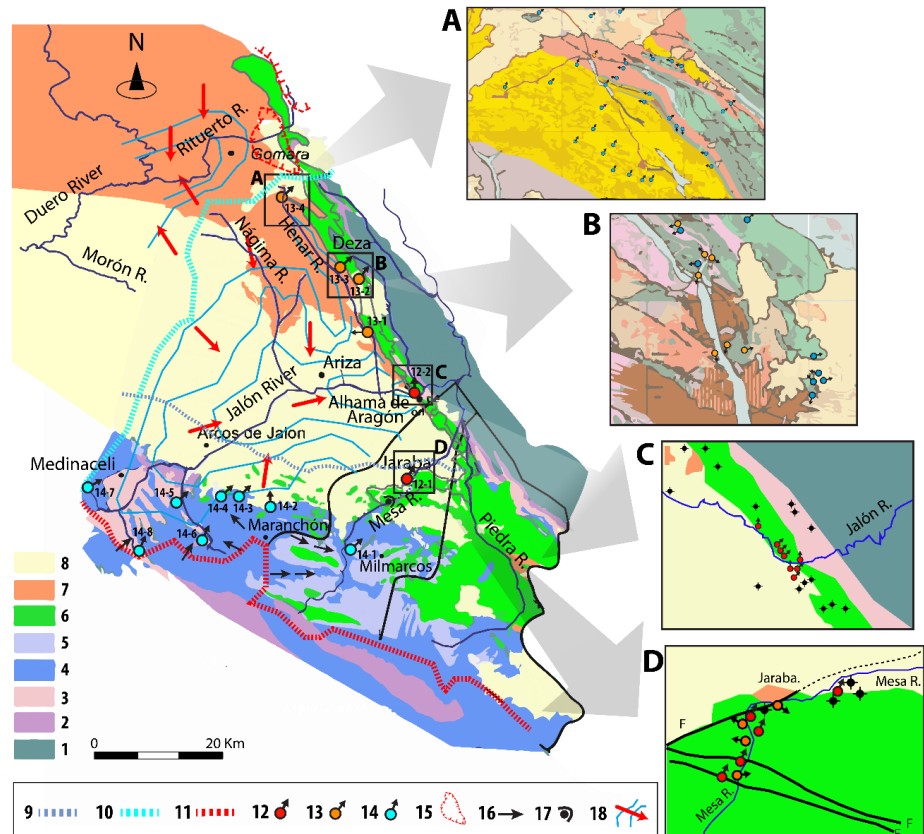

**Fig. 3.** *Hydrogeological diagram of the study area.*

*Geology: 1. Precambrian and Palaeozoic: quartzites and shales. 2. Lower and Middle Triassic: sandstones of the Buntsanstein facies and dolomites and marls of the Muchelcalk facies. 3. Upper Triassic: clays and gypsum of the Keuper facies. 4. Lower Jurassic: Carniolas (dolomites with small cavities) and dolomites. 5. Middle and Upper Jurassic: limestones and marls. 6. Cretaceous: Utrillas facies sands below. Upper limestones and marls. 7. Paleogene of the Northern Zone (adapted from Huerta, 2007). 8. Neogene: shales, siltstones, conglomerate. 9. Jurassic boundary under the Cenozoic.*

*Hydrography and hydrogeology: 10. Water divide between the Duero-Ebro. 11. Water divide between the Ebro-Tajo. 12. Group of thermal springs (12.1 Jaraba. 12.2 Alhama de Aragon). 13. Group of semi-thermal springs (13.1. Embid de Ariza. 13.2. San Roquillo. 13.3. Deza. 13.4. Almazul). 14. Important cold springs in the Sierra del Solorio (14. 1. Mochales. 14.2.Iruecha. 14.3. Chaorna. 14.4. Sagides. 14.5. Urex. 14.6. Layna. 14.7. Ambrona. 14.8. Esteras de Medinaceli or source of the river Jalón). 15. Poljes of the river Rituerto. 16. Flow lines in the Sierra del Solorio. 17. Sinkholes in the river Mesa. 18. Ground water contour and surface flow lines in the Tertiary of the Almazan Basin.*

*Table A. Detailed location of springs in the Almazul area associated with the Palaeogene and Cretaceous (geological base taken from Huerta, 2007).*

*Table B. Detail of the location of the group of springs in the Cretaceous calcareous aquifer in Deza and San Roquillo, differentiating between deep flow (orange) and shallow flow (blue) (geological base taken from Huerta, 2007).*

*Table C. Detail of the situation of the group of thermal springs (red) and boreholes of the Cretaceous-calcareous and Tertiary aquifer (T) in Alhama de Aragón.*

*Table D. Detail of the situation of the group of thermal springs (red), cold springs (blue) and boreholes of the Cretaceous calcareous and Tertiary (T) aquifer in Jaraba area.*





### 2.3.2.1 *Karstification of Cretaceous limestones in the Cretaceous-Tertiary discontinuity: a source of extensive and deep porosity for the thermal aquifer*

In this area, apart from the karstification developed during the Neogene-Quaternary, as the most determining period for the current or sub-current morphology, traces of older karstification can also be observed in the Cretaceous limestones, as well as in the Cretaceous-Tertiary sedimentary discontinuity, which holds special hydrogeological importance in this thermal aquifer.

The presence of signs of karstification along the Cretaceous-Tertiary discontinuity at the border of the Almazán Basin with the Aragonese branch is a general feature identified by several authors (Hernández-Pacheco, 1954, Armenteros, 1989, Huerta, 2007). In addition, all the thermal or semi-thermal springs are located in this contact of different permeability, so the relationship between both facts is evident, and the interest in characterizing this source of porosity. In this karstification, we must distinguish the labyrinthine conduits in the limestones in contact with the Tertiary generated by the circulation of groundwater in the thermal aquifer during the Neogene-Quaternary (Hernández-Pacheco, 1954), which can be considered to be a consequence of the circulation of the flow according to a scheme more or less similar to the present one (Fig. 2). And another much older one, prior to the configuration of the aquifer, and which is represented by the paleokarsts developed in the Paleocene and Lower Eocene during the exposure to the exterior of the Upper Cretaceous limestones (Armenteros, 1989, Huerta, 2007), and which probably constitute the main porosity at the depth of the thermal aquifer. It is not excluded that the more modern conduits in the upper part of the aquifer have taken advantage of this original karstification.

This karstification is recognized in at least seven points distributed between Deza (Soria) and Godojos (Zaragoza) in the Cretaceous limestones (Armenteros, 1989), but there are probably many more sites (Fig. 2). The fact that these points are spread over a length of 30 km and are distributed at different altitudes according to a maximum difference in altitude of 350 m, indicates that karstification is quite widespread. Then, during the subsequent major sedimentation, this karstified discontinuity was buried and submerged by subsidence to a depth of more than 4,500 m. This provided a widespread source of porosity at great depth, with geothermal gradients of around 100 ºC, which may have allowed large flows of water to penetrate, heat up, and circulate along the basin floor. The degree and intensity of this karstification is unknown, but it is thought to be unimportant as the porosity values required in the aquifer modelling have not been large (0.3 %).

### 2.3.2.2 *Geometry of the Thermal Aquifer and Deep Structure of the Almazán Basin*

On this northern edge of the Aragonese Branch at the contact with the Almazán Basin, the structure has a general NW-SE orientation, unlike the edge with the Castilian branch, which is dominated by a W-E direction. At depth and based on the ground water contour of the Tertiary base obtained from reflection seismic surveys, oil drilling, and the support of gravimetric studies, the geometry at depth of the Almazán Basin, and specifically of the Cretaceous calcareous aquifer, can be deduced (Maestro, 2004). The structure of the basin is characterized by folds with NW-SE to E-W directions, associated in some cases with north-vergence thrusts. Two sectors are considered (Fig. 4): to the west of the Almazán-Soria meridian, the fold axes are E-W, while to the east, as they approach the Aragonese branch, they acquire NW-SE directions typical of this





unit of the Iberian Range. In the eastern sector of this basin, three areas with different structures can be

distinguished (Fig. 4):

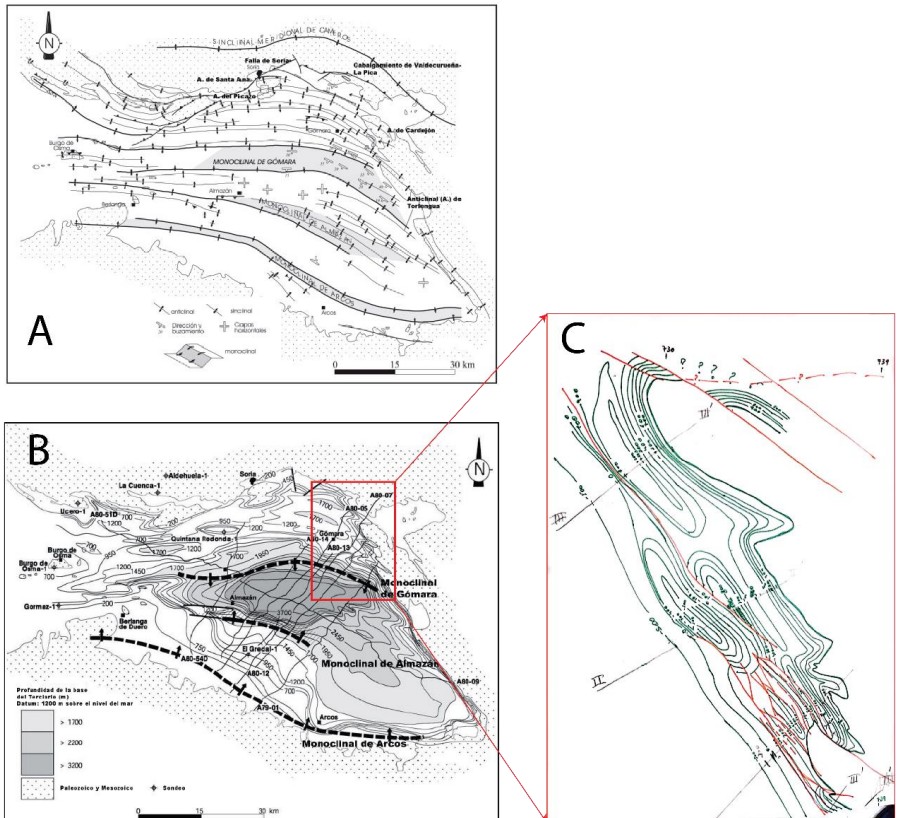

**Fig. 4.** *Geometry in depth of the Cretaceous-thermal aquifer. A. Location map of tectonic structures present in the Almazán Basin (taken from Maestro, 2004). B. Contour map of the Tertiary base in the Almazán Basin made from seismic profiles (taken from Maestro and Casas Sainz, 2000). C. Detail of the*
*structural contour isolines of the base of the Cretaceous thermal calcareous aquifer at a rim of the Aragonese branch acco   rding to the bounded plane method.*

In the center of the basin, east of Almazán, the main structure is the bottom of a syncline bordered by the Gómara monocline to the north, and the Almazán monocline to the south.

The Gómara monocline dips 20º-30º to the south, gradually decreasing to the west. The folds are slightly
asymmetric with north vergence and are associated in many cases with faults affecting the basement (Maestro, 2004; Casas-Sainz et al., 2000).

To the south is the Almazán monocline, which has very low northward dips in this eastern part, and further south the Arcos monocline dips in the same direction (Fig. 4). North of the Arcos monocline there is no more Jurassic, and this is observed in the seismic profiles from the Soria meridian to the end of the Almazán Basin
at Jaraba, as well as in the Gredal-1 survey. On the southern flank of this monocline, the Cretaceous and especially the Jurassic rocks of the Castilian branch outcrop.



## 3    Methodology

### 3.1    Construction of the conceptual model

Firstly, a complete bibliographic review was carried out, followed by intensive fieldwork, compiling as much
information as possible on the system under study in order to construct a conceptual model that formed the
basis for the development of the numerical modelling. In this model, the geometry of the Cretaceous
carbonate hydrothermal aquifer has been graphically represented, considering the surface topography,
geology, main hydrogeological units and their contacts, existing wells, and the extension of the sector to be
modelled. By integrating different sources of information and applying interdisciplinary methods, sufficient
knowledge of the thermal system has been obtained for its modelling. Specifically, the methodology followed
is as follows:

- **Palaeogene stratigraphy and origin of the karstification of the hydrothermal aquifer:** In this respect,
   previous studies in the area were analyzed, including the work of Armenteros (1989) and Huerta (2007),
   in order to determine the shape, distribution and extent of the Palaeogene units and the origin of the
karstification in the Cretaceous limestones, which directly influence the hydraulic conductivity of the
   hydrothermal aquifer.

- **Geometry of the hydrothermal aquifer and application of the bounded plane representation
   technique:** This knowledge of the geometry will help to understand the geological structure and the
   existence of possible tectonic features that could break the continuity of the Cretaceous series that in
principle is assumed, and that could modify the underground flow. Furthermore, in cases such as this one,
   and in accordance with the regional tectonic context, only a normal geothermal gradient is available, so
   that the identification of the heat source that gives rise to the geothermal system will be the areas of the
   aquifer that reach the greatest depth.

In order to know the extent and limits of the Cretaceous water-bearing materials, the work of Maestro
(2004) (Fig. 4)

However, although reflection seismic and gravimetry provided the geometry of the deep parts of the basin,
this map was too general for the edges of the basin, which is where the discharge zones of the Cretaceous
calcareous aquifer are located, which are key in the process of simulating groundwater flow, for example.
In addition, apart from the surface geology, data from other groundwater abstraction boreholes in the
Tertiary have been used, which have reached the Cretaceous and Jurassic (CHE, 2023), as well as
geophysical studies in the Domain of Time, SEV and gravimetry around Gómara and Almenar de Soria
(JJCCLLL, 1991). All this has been used to draw the structural contour lines of the Tertiary base by the
method of boundary representation systems in the aforementioned edge zones, linking them with those of
the deeper parts of the Maestro (2004) map.

- **Hydrothermal system domain boundaries:** how the Cretaceous hydrothermal aquifer continues and
   extends into the Duero Basin, in order to locate the hydrogeological divide of the hydrothermal system
   that drains the springs located in the Ebro Basin of Alhama, Jaraba, The results of several works and





doctoral theses on the Cretaceous hydrogeology of the Almazán Basin rims in the Duero Basin, among which the following stand out: Pérez Santos (2007), Távara Espinoza (2011), and Sanz et al, (2022).

- **Inventory of points:** from the information of the inventory of water points and pumping tests in wells, specific flow rates and ranges were determined for the permeability values of the different hydro-stratigraphic units, as starting values to support the mathematical model. The inventory in the Cretaceous-Tertiary aquifer is scarce and has not allowed groundwater contour maps to be drawn up, but it has been important for identifying different types of flow and discharge and recharge zones. In the Tertiary, the

inventory has served to confirm its low permeability and the identification of aquitards in the Palaeogene.

An inventory of 540 water points has thus been made, including permanent or ephemeral springs (70 %), boreholes and wells (28%), galleries and effluent streams (2%). As the study area is very sparsely populated, with one of the lowest population densities in the European Economic Community, there are hardly any boreholes that could be used for a general isopieces map. Among the boreholes, there are 3

piezometers of the piezometric control network of the "Confederaciones Hidrográficas del Ebro" (CHE, 2023) and "del Duero" (CHD, 2023), which have been used to calibrate the groundwater flow simulation.

    - **Recharge calculation:** Although the modelling has made it possible to estimate recharge in detail, the previous evaluations of Sanz and Yelamos (1998) based on the hydrometeorological records of the study area were used as indicative starting values. These evaluations have been modified, mainly taking into

account the variation of rainfall and evapotranspiration with altitude and the different topographical characteristics of the Jalón and Duero basin.

    - **Isotopic Hydrogeochemistry:** To improve the hydrogeological and hydrogeochemical conceptual model. A tritium isotopic monitoring of the thermal and semi-thermal springs has been carried out to improve the understanding of the groundwater dynamics in the system and to clarify the origin and age

of the Alhama and Jaraba springs. All the above information has been compiled (Taken from ITGE-DGA (1994), Yélamos and Sanz (1998) and several sampling campaigns have been carried out over the last two decades. Samples have also been taken in these springs for environmental isotope analysis (Deuterium and Oxygen 18) in an effort to determine the area of origin of the natural recharge.

### 3.2  Modelling of hydrothermal system flow

The hydrogeological conceptual model is known, and the subsurface flow situation has been simulated with MODFLOW. At a regional or basin scale, it is customary to assume that the rock medium behaves as an equivalent porous medium with blocks whose permeabilities adequately reflect the hydraulic behaviour of the flow in the system. This modelling has allowed us to test and refine the previous hydrogeological conceptual model. The model was subsequently calibrated in both steady and transient regimes and then

subjected to a sensitivity analysis of the main parameters conditioning the behaviour of the system.





## 4   Results and discussion

### 4.1   Tertiary hydrogeology

#### 4.1.1   Aquitards identification.

From the inventory of water points carried out, specific flow data from 60 boreholes and wells were available,
as shown in Table 1. Although not all of them come from detailed pumping tests, most of them are considered
to be quite reliable. Classified by lithological groups of similar hydrogeological behaviour, 17 correspond to
the alluvial of the Jalón river and also of some tributaries, 23 to the Miocene (except the limestones of the
Páramo), 1 to the Ocino Fm. of the Palaeogene, 1 to the Gómara Fm. of the Palaeogene, 9 to the limestones
of the Upper Cretaceous, 8 to the Lías dolomites (Jurassic), and 1 to the Muchelkalck dolomites (Table 1).
Transmissivity has been obtained according to the approximate formula of Calofre (Custodio and Lamas,
1975) and permeability considering the saturated thickness of the wells and transmissivity.

**Table 1.**   *Specific flow, transmissivity, and permeability values of some of the hydro stratigraphic formations in the study area*

| Lithological group | Nº de pumping test | Specific flow rate $q_e$(l/s/m) | Transmissivity T (m²/day) T=100 $q_e$ | Permeability K=T/b (m/day) |
|---|---|---|---|---|
| Aluvial deposits | 17 | 1,47 | 147 | 30 |
| Miocene period | 23 | 0,082 | 8,2 | 0,1 |
| Paleogene period (Fm.Gomara) | 1 | 0,105 | 10,5 | 0,06 |
| Paleogene period (Fm. Ocino) | 1 | 0,2 | 20 | 0,5 |
| Cretaceous | 9 | 8,3 | 830 | 6-11,8 |
| Jurassic (Lias) | 8 | 0,56 | 54 | 1,14 |
| Muchelkalk | 1 | 0,012 | 1,2 | 0,024 |

As far as the Tertiary is concerned, and based on these data, the detailed knowledge of the lithology, as well
as the distribution and flow of springs and in this area, it can be concluded that both the Palaeogene and the
Neogene of the study area are, overall, not very permeable.

The Tertiary of the Almazán Basin of the Duero hydrographic basin has been considered as an aquifer-
aquitard of low permeability (officially called the Tertiary Aquifer of southeast Soria (IGME, 1983). For the
CHE (2023), this same Tertiary has been considered impermeable and, in the Ebro Basin, (Upper Jalón) and
no type of aquifer has been defined.

#### 4.1.2   Two Tertiary flows: shallow and deep.

-   **Surface flow:** Fig. 3 presents a schematic Tertiary water table contour map based on 420 water points
    and rivers/streams flowing with water. The qualitative interpretation of this map indicates that: 1. The
    water table contour is quite adapted to the topography, with the water level of the boreholes close to the
surface, which is typical for low-permeability soils. The impermeability of the geological formations of
    the basins of the rivers Nágima, Santa Cristina and Jalón, for example, has been known for a long time





and this has limited human supply by means of wells (Mendizábal and Cincunegui, 1941, SGOP; 1984).
2. Also as a consequence of the above, the Duero-Jalón surface and underground divides coincide at
surface level in most of the study area, except for the divide with the Rituerto river. 3. The rivers and
streams of the Jalón Basin drain, but the discharge is not continuous and diffuse, but occurs through small,
occasional springs which are associated with more permeable levels within the Tertiary, either Neogene
or Palaeogene. Some of these springs are located near the watershed. Most of the rivers do not discharge
significant groundwater; this can be seen very well in the dry season when a large part of the riverbeds
run dry, such as the Jalón in droughts or the Nágima.

On the other hand, as shown in Fig. 3, we observe the large number of facies and lateral facies changes
within the Palaeogene, which makes hydraulic continuity between the Tertiary aquitards and the
Cretaceous difficult, since they are almost always surrounded by impermeable formations. This favours
the disconnection and hydraulic isolation of the Palaeogene geological facies, and the subsurface flow is
predominantly superficial and of limited spatial extent. Therefore, there is a relatively high number of
small springs draining more permeable, weakly permeable layers interspersed between a mass of
impermeable sediments (Fig 3; Detail A).

- **Deep flow:** The Rituerto river and its tributary the Arroyo de la Vega in the Duero Basin have the
behaviour of losing rivers, even though they flow through Tertiary sediments. Despite having a large
catchment area (more than 600 km$^2$), it is usual to see them dry most of the year (hence the historical
name of Tuerto River, for example, which refers to a half-dry river). This is because the Tertiary is not
very thick here and the Cretaceous calcareous aquifer is close to the surface, sometimes outcropping, and
this absorbs the recharge. All this has led to the formation of two large karstic depressions or poljes filled
with recent Tertiary and Quaternary sediments: the Rituerto polje, and the Cañada Hermosa karstic
depression (Echeverría, 1989). This Cretaceous proximity to the surface has been geometrized in detail
by means of different geophysical techniques in the work of (JJCCLL, 1991).

Thus, it can therefore be concluded that two flows coexist in the Tertiary of the area, a sub horizontal
surface flow when the Tertiary is strong, and a deep flow when the underlying Cretaceous calcareous
aquifer is nearby. This flow, with a significant vertical component, feeds the thermal aquifer. Actually,
what it can be observed in Rituerto and Arroyo de la Vega is the surface manifestation of what is
happening at depth: a zone of influence or absorption of Tertiary groundwater by the more permeable
Cretaceous aquifer below (Fig. 5).



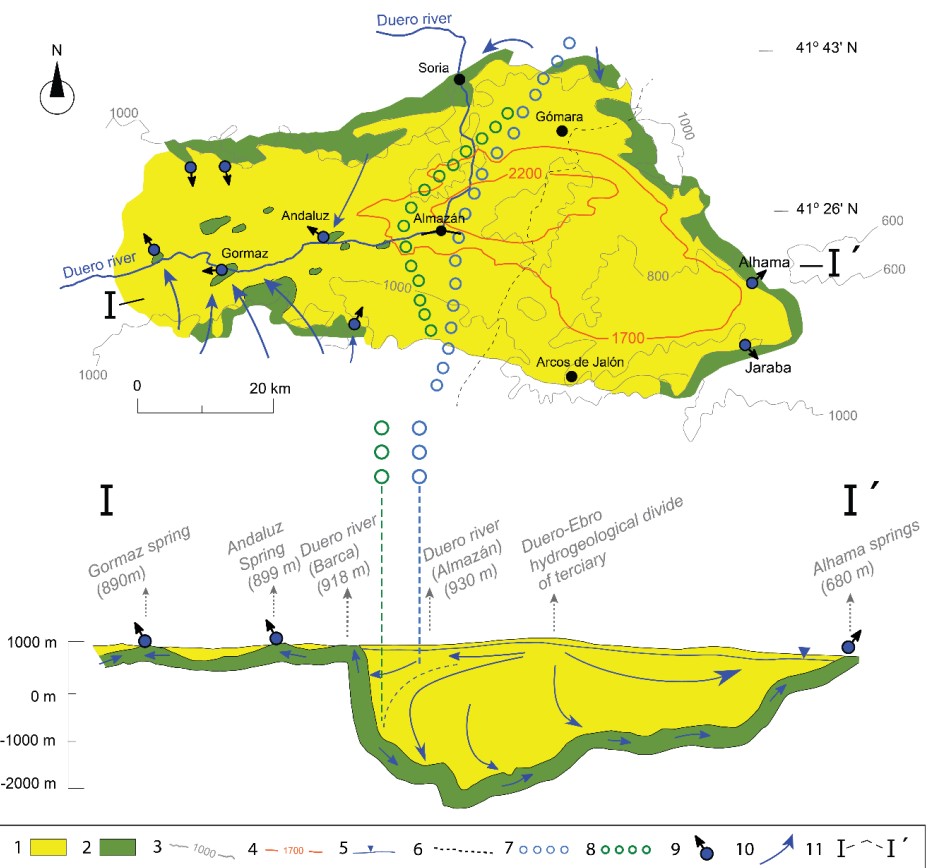

**Fig. 5.** *Schematic hydrogeological cross-section along the bottom of the Almazán Basin showing the hydrogeological divide established in the mathematical model for the Cretaceous-Thermal aquifer.*

*1. Topographic contour lines. 2. Duero-Ebro surface divide. 3. Edges of the Almazán Basin (mainly carbonate aquifer). 4. Tertiary of the Almazán Basin. 5. Depth of the top of the Cretaceous-thermal aquifer. Hydrogeological cut (the power of the Cretaceous-thermal calcareous aquifer is exaggerated) 7. Water table 8. Springs. 9. Flow lines. 10. Duero-Ebro hydrogeological divide, verified at the edges and assumed in the interior of the basin. 11. Duero-Ebro hydrogeological divide, maximum assumed position.*

*4.2    Hydrogeology of thermal aquifer*

*4.2.1    Semi-thermal and thermal manifestations in the Cretaceous of the Almazán Basin.*

Fig. 1(B) is a geological and topographical diagram showing the Almazán Tertiary Basin as a whole, including the area belonging to the Duero Basin. In this basin, the topography ranges between 900 and 1000 m for the most part, while in the Ebro basin (Jalón river), the altitudes range from 640 m in Alhama de Aragón

to 1,100 m at the boundary with the Duero. The highest peripheral areas correspond to the Mesozoic calcareous rims, while the lower central areas are occupied by Tertiary terrigenous sediments. Also noted are those springs that have a temperature 4ºC above the average of the rest of the springs in the region, which is around 11ºC, and which range between 15ºC and 18ºC in the Duero Basin (semi-thermal springs), and between almost 20ºC and 32ºC in the Ebro Basin (thermal springs).





In Fig. 1, 3 and 5 we can see how the semi-thermal and thermal springs in the Almazán basin are always associated with the calcareous Cretaceous and are found at the contact between this Cretaceous and the Tertiary. In the Jalón basin all the springs range between 18ºC and 32ºC and are located on the NE edge, except for Jaraba, which is nearby; in the rest of the southern peripheral edge, regardless of whether the Tertiary contact with the Cretaceous or Jurassic, there are no other manifestations. This Cretaceous-Tertiary

contact has a drop of more than 400 m between the Duero-Ebro divide and Alhama de Aragón (Fig. 5). In the Duero these semi-thermal manifestations are usually located in the lower altitude anticlinal domes that emerge from the Tertiary, and represent the emergence of medium depth flows, since in the Soria plateau there are no significant topographic differences between the recharge and discharge areas (Pérez Santos, 2007; Távara Espinoza, 2011 and Sanz et al, (2022).

*4.2.2    Tectonic fractures that break the hydraulic continuity of the Cretaceous-Thermal Calcareous aquifer at depth.*

As mentioned in section 2.3.2.4, this work follows the interpretation of Maestro (2004), on the deep geological structure of the Almazán Basin. This assumes that the Upper Cretaceous thermal aquifer has continuity and is hydraulically connected to each other under the Tertiary. However, in certain sectors,

different interpretations have to be considered, such as that of Bond (1996), where he admits that the Cretaceous lateral continuity is sometimes broken by reverse faults with important dips. This is also observed in the surface geology, such as the apparent interruption of the Cardejón mountains with the Castejón mountain range through the Soria fault, for example. The identification of tectonic fractures that locally interrupt the hydraulic continuity within the aquifer can be important for the modelling of the flow. In order

to have the most realistic geometry possible, the previous information from the above-mentioned authors has been complemented by the dimensioning representation system. According to this analysis, we have been able to identify two features that have been considered in the modelling phase:

The first one refers to the sector between Embid de Ariza and Alhama. It is a series of several NW-SE isopach anticlines with westward vergence, involving the Palaeogene to the Triassic and faulted by subvertical reverse

faults involving the Palaeozoic. These folds have verticalized western flanks and the eastern flank with lower dips. Toward the center of the basin, there is a series of N-vergence folds, of which only the easternmost one is at the surface (east end of the Fuentelmonge Anticline), which has an ESE direction, oblique to the general structure of the Aragonese branch.

Deformation is progressively attenuated toward the basin center and toward deposits of more modern age.

The most important implication from a hydrogeological point of view is that the fault jumps associated with some of these fault-folds displace the impermeable Triassic and Palaeozoic substrate in such a way that they act as a vertical barrier, interrupting the continuity of the Cretaceous south of Deza with the Embid area.

The second structure with hydrogeological implications is the Jaraba fault (Lendinez and Martin, 1991), a fault with a NE-SW direction and a large dip, dipping the NW block and confronting the impermeable Triassic

and Mesozoic rocks with the Upper Cretaceous. The result is an impermeable barrier effect that isolates the area furthest from the periclinal closure of the SE Ibdes from the rest of the thermal aquifer.



All these hydrogeological disconnections have been simulated in the numerical model and as indicated, play a fundamental role in the calibration of this model.

### 4.3    Simulation of groundwater flow in the thermal aquifer

*4.3.1    Code used.*

The study area can be represented by a three-dimensional finite difference grid, on which the system of differential equations of groundwater flow can be solved.

The mathematical code used is Modflow 2005, a numerical code that uses the three-dimensional finite difference method to simulate flow in the saturated zone. The flow regime can be stationary or transient. The
Modflow-2005 version of Model Muse is used as the graphical analysis interface.

### 4.3.2    Location, boundaries, and geometry of the model domain

According to the above, the domain of the hydrogeological model of the Cretaceous hydrothermal aquifer covers the entire Tertiary of the Almazán Basin between the hydrogeological divide of the Ebro to the peripheral Cretaceous edges of the Aragonese branch of the Iberian Range, and the Jurassic-Cretaceous of
the Castilian branch. In addition, the domain of the model area is extended to the West, entering the Duero Basin. The hydrogeological divide of the hydrothermal aquifer does not coincide with the Duero-Ebro hydrographic divide, as justified above. The extension and geometry of the model are shown in Fig. 6.

The area modelled in the model covers an area of approximately 2295.5 km$^2$

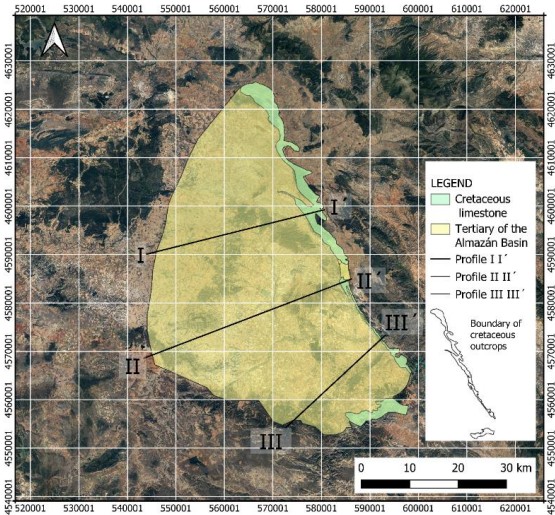

**Fig. 6.** *Domain of the hydrothermal aquifer (© Google Earth.)*


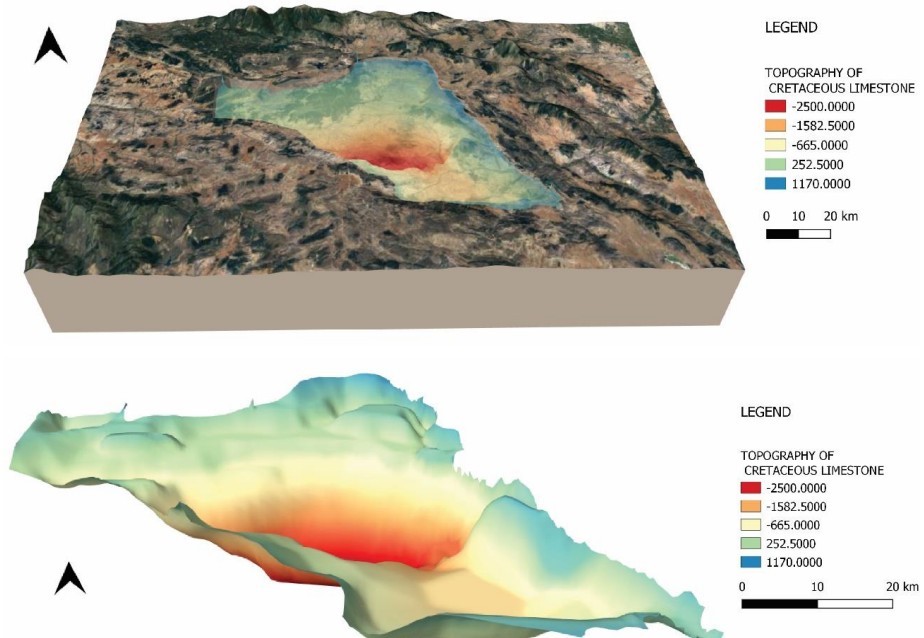

**Fig. 7.** *3D image of the Cretaceous hydrothermal aquifer geometry (© Google Earth)*

### 4.3.3    Meshing and discretization

The model considers two differentiated horizontal layers: Layer 1. Upper layer of sedimentary deposits corresponding to the Tertiary that constitutes the Almazán basin. Mainly formed by Miocene and Palaeogene. Low permeability layer. Variable thickness, ranging from a few meters in the outcrop area to 4000 meters in the central area of the basin; and Layer 2. Lower layer constituting the Cretaceous limestone aquifer, which is confined by the Tertiary. Average thickness of about 300 meters, except in the outcrops and surrounding areas, where greater apparent thicknesses can be observed, reaching up to 800 meters in some cases.

Both layers have been discretized into three horizontal layers in order to improve the calculation of the programme, as seen in Fig. 9, Fig.10 and Fig.11.

The mesh used in this model is of the cell-centered type. The overall dimensions of each cell in both layers are 500 × 500 m. In the areas of greatest interest, such as the edge of outcrops where springs are located or areas with a high density of contour lines, a spatial discretization (mesh refinement) has been carried out to achieve greater accuracy in the simulation. The model is made up of a mesh of 49621 cells, as shown in Fig 8.

The roof and wall elevations of each layer are entered, matching the geometrical model as far as possible with the morphology of the geological contacts. Thickness is variable in both layers.

- The roof of layer 1 reflects the topography.





- The roof and base of layer 2, as indicated in chapter 2, is also of variable thickness and has been
defined by geophysics and by the dimensioning representation system.

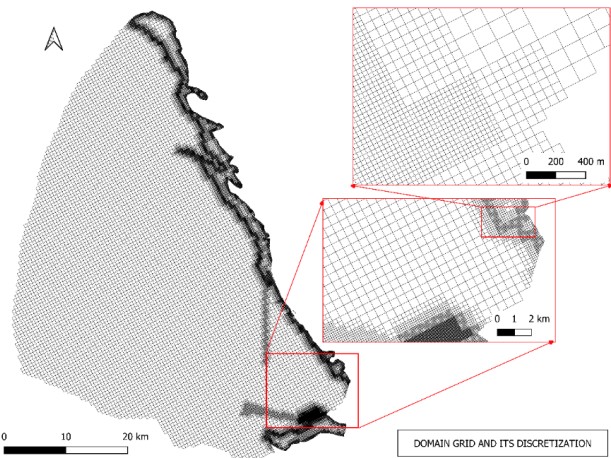

**Fig. 8.** *Meshing and mesh discretization*

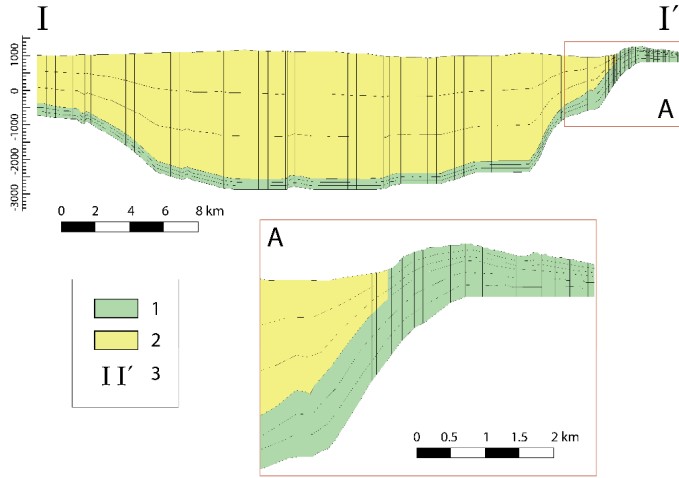

**Fig. 9.** *Layers of the hydrogeological model. Profile I-I´.*





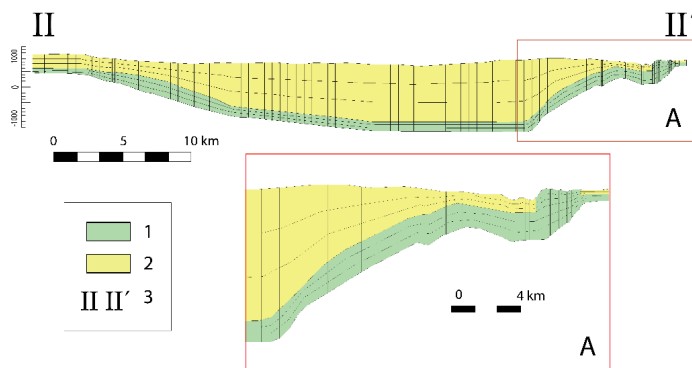


**Fig. 10.** *Layers of the hydrogeological model. Profile II-II´.*

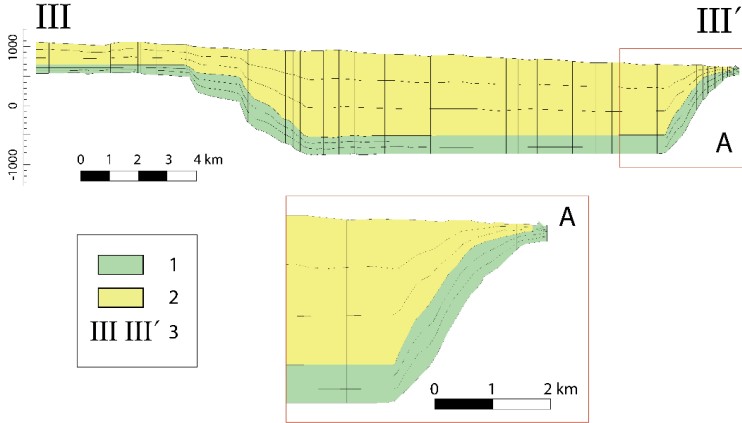

**Fig. 11.** *Layers of the hydrogeological model. Profile III-III´.*

*4.3.4 Hydrogeological parameters*

*4.3.4.1 Hydraulic permeability*

The ranges of hydraulic permeability values established in the model are based on the initial horizontal hydraulic permeability values taken from the pumping tests (Table 1). These initial values have been adjusted to calibrate the numerical model. It should be noted that during the calibration period the following geological and hydrogeological considerations have been considered, which have been fundamental in successfully 650 calibrating the mathematical model.

- According to Darcy's Law, the hydraulic conductivity is not characteristic of the porous medium, but also depends on the fluid.

$$K = k\frac{\gamma}{\mu}$$

Where, K is the hydraulic conductivity, k is the intrinsic permeability (dependent only on the porous 655 medium), γ the specific gravity of the fluid, and μ the dynamic viscosity of the fluid. It should be





noted that only the variation of viscosity with temperature should be considered (Custodio and Llamas, 1975). In general, groundwaters show minimal temperature differences throughout the year in the same aquifer, but in other environments there can be notable temperature differences, as in this case study.

The base of the Cretaceous ceiling at the bottom of the basin reaches depths of around 4000 meters. Considering the natural geothermal gradient, which indicates an increase of 3º C for every 100 meters of depth, and assuming an average water infiltration temperature of around 10-12ºC, we can assume that in the deepest part of the Cretaceous limestone aquifer the water would be at an average temperature of between 100-120ºC.

According to the properties of water, a higher temperature produces a reduction in viscosity and therefore, as indicated above, an increase in hydraulic conductivity. This translates in the model into a zonal variation of permeability, depending on the depth, as shown in Fig. 12.

- The system would be practically equivalent to an anisotropic homogeneous aquifer with a vertical permeability smaller than the horizontal permeability according to Llamas and Cruces de Abía,
1976. Therefore, in general, a vertical permeability 10 times smaller than the horizontal permeability ($Kz = kxy/10$) has been defined in the model. This is considered when the strata are horizontal, but as the Cretaceous limestones reach the outcrops of the Aragonese branch, the strata become vertical and therefore the permeabilities kx, Ky and Kz have to be redefined, so that the axes of the strata coincide.

Therefore, considering the initial values described in the conceptual model and taking into account the considerations described here, the following hydraulic conductivities have been calibrated in the model. Hydraulic conductivities that fit very well with those defined in the conceptual model.

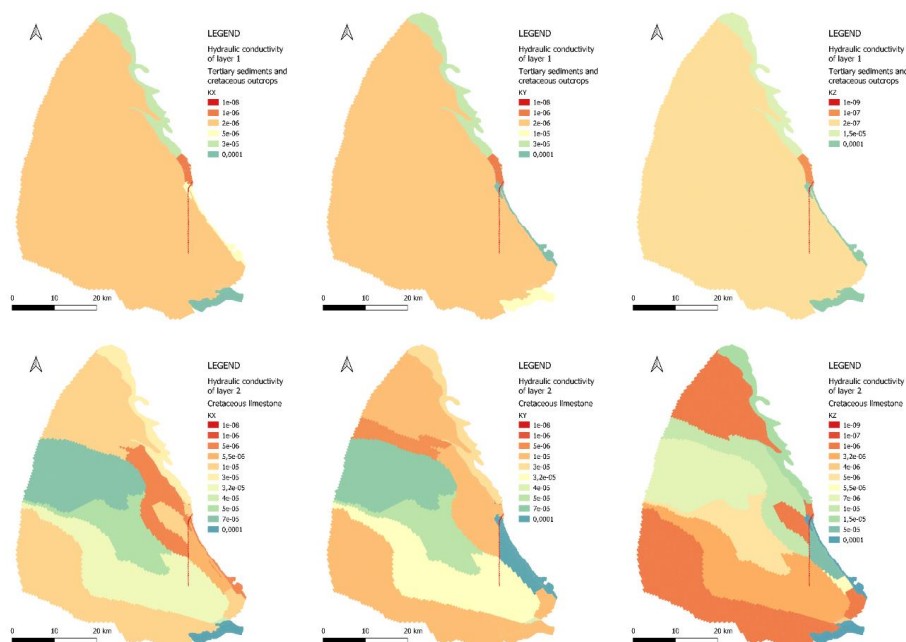

**Fig. 12.** *Distribution of the permeabilities Kx, Ky & Kz in the mathematical model*

*4.3.5    Outline conditions*

According to the conceptual model, four types of contour conditions have been defined in the model.

*4.3.5.1    Recharge*

There are two types of recharge of water by effective infiltration. The first corresponds to direct or autogenous recharge from precipitation recharge. Secondly, there is an indirect or diffuse recharge, coming from the infiltration of the Mesa River as it passes through the Cretaceous limestone between the village of Calmarza

and Jaraba, as mentioned above.

The autogenous recharge of precipitation in the field has been simulated in the model using the MoD Flow recharge package (RECHARGE).

As the study area is relatively large, around 2295.5 km², a spatial distribution of recharge has been considered. As initial and indicative values, the previous assessments of Sanz and Yelamos (1998) based on the

hydrometeorological records of the study area were used as starting values.

As shown in Fig. 13, the annual recharge is differentiated into different recharge areas according to the initial values and the following considerations. This recharge corresponds to each complete hydrological year modelled in the stationary period.

However, due to the existence of precipitation and evapotranspiration gradients depending on altitude, the

greatest volume of water infiltrates at the western end of the limestone band.





- As previously indicated, the higher the altitude, the higher the precipitation. Therefore, the northernmost areas of the model, being at a higher altitude, contribute to a higher recharge to the model.

- In addition, there are two poljes of special recharge. The first is Villaseca de Arciel and the head of the Rituerto (considered to be part of the recharge of the Sierra de la Pica and Sierra de Tajahuerce).

- Recharge in the Tertiary of the Duero basin has been considered in the model to be higher than in the Ebro basin: In the Duero basin there are flat endorheic areas with lower slopes, which favours recharge and reduces surface runoff. There are also two poljes with periodic flooding.

The direct or allogenic recharge through drains in the bed of the Mesa River as it passes through Calmarza
has been simulated using the "WELL" package, which allows us to consider a flow of water leaving or entering a given cell. This water inflow has been taken according to the values of (IGME, 1980, 1987) where it is indicated that this recharge is mainly visualized during low water periods and is of the order of 0.100 $m^3$/s to 0.4 $m^3$/s.

Therefore, a total allogenic recharge rate of 0.2 $m^3$/s has been considered, distributed homogeneously along
the course of the river Mesa between the villages of Calmarza and Jaraba. This recharge rate has also been calibrated by the numerical model.

Once the model has been calibrated, the final estimated and calibrated recharge is as shown in Fig. 14.

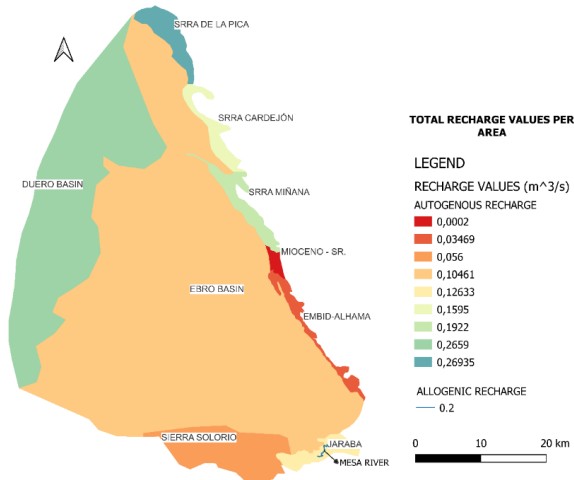

**Fig. 13.** *Spatial distribution of diffuse and allogenic recharge values*

*4.3.5.2    Springs*

As has been said, the outflows associated with this model correspond to the discharge areas of the water from the Cretaceous limestone aquifer, which are produced through the high-flow Alhama de Aragón and Jaraba springs, as well as through other springs associated with the same geological area, which have medium to low flows.





These springs have been simulated in the numerical model using the Modflow drainage package (DRAIN).

The drainage package has also been simulated along the outcrop edges of the Cretaceous limestone outcrops in the areas where the springs occur, to be able to have a greater degree of detail and not only limit it to the cell where the springs are located. In this way, a better ability to calibrate the piezometric level is achieved. The Drain boundary conditions are shown in detail in Fig. 14.

*4.3.5.3    Edge conditions. No-flow.*

The definition of this behaviour is carried out through the boundary conditions imposed on the model. The following boundaries have been considered closed in the model.

- Jaraba fault. As explained in section 4.2.2., the Jaraba fault produces a disconnection of the Cretaceous limestones in a south-easterly direction. This disconnection justifies why there are no
thermal water springs in the outcrops of Cretaceous limestone in the town of Ibdes, as these outcrops are at a lower altitude than the Jaraba springs. This edge has been modelled as a no-flow zone.
- Impermeable boundary to the west of the Cretaceous limestone outcrops, corresponding to the end of the Cretaceous limestone aquifer. This boundary has been modelled with a no-flow zone.
- Hydrogeological divide boundary to the east of the model. Non-zero flow boundary.
- Impermeable boundary to the south, corresponding to the contact with the Sierra del Solorio.

*4.3.5.4    Hydrogeological disconnections.*

The Cretaceous along the edge of the Cretaceous limestones of the Aragonese Branch as it passes through Embid and extending southwards, is disconnected by a series of thrusts through which the impermeable materials of the Mesozoic and Palaeozoic substratum have been uplifted. In some cases, these fault jumps
cause barriers, and a hydraulic disconnection of part of the aquifer. The existence of these thrusts has been fundamental in calibrating the model. Firstly, it has created a quasi-impermeable barrier causing isolation of the Embid de Ariza outcrops. These thrusts have created a disconnection of the preferential flow of water from the outcrops of La Pica and Cardejon toward Alhama, thus causing a diversion of the flow from the north toward the centre of the basin, which would justify the heating of the water. Even so, due to the high
hydraulic pressure, a certain amount of water seeps through the thrusts, which justifies the existence of the 20 l/s of the Embid of Ariza springs. Therefore, this boundary condition has been modelled as a low permeability zone, which has been calibrated.

The boundary conditions are shown in Fig. 14 below.





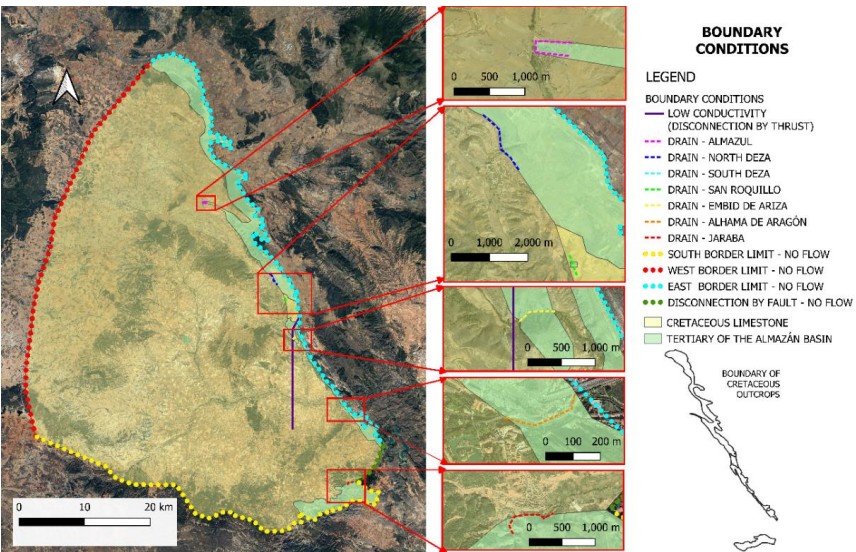

**Fig. 14.** *Drainage boundary conditions, low permeability and no-flow (© Google Earth.).*

*4.3.6    Simulation and model calibration*

Calibration is the process of modifying the input parameters in the model until the solutions fit and match the observed data.

The objective is to solve the so-called Inverse Problem, i.e., trying to estimate and adjust the system
parameters (such as conductivity, specific storage and recharge) in such a way that the model is able to reflect the real behaviour of the groundwater recorded at the site from direct measurements of both spring flows and observation points (piezometric levels).

Firstly, it is worth highlighting the difficulty of calibrating this mathematical model. The great extension of the model (2295.5 km$^2$), the geological complexity, the great variety of values of hydraulic conductivities,
the high number of boundary conditions, etc., which together with the scarce hydrogeological knowledge and scarcity of piezometers has meant a great difficulty when adjusting the mathematical model. In any case, the numerical model was successfully calibrated in accordance with the conceptual model.

Mathematical model calibration was based on the adjustment of the piezometric levels and the flow rates of the successive thermal upwellings of the springs.

The simulation of the numerical model results in the groundwater contour shown in Fig. 15 where the state of the piezometric level of the Cretaceous limestone aquifer is observed.



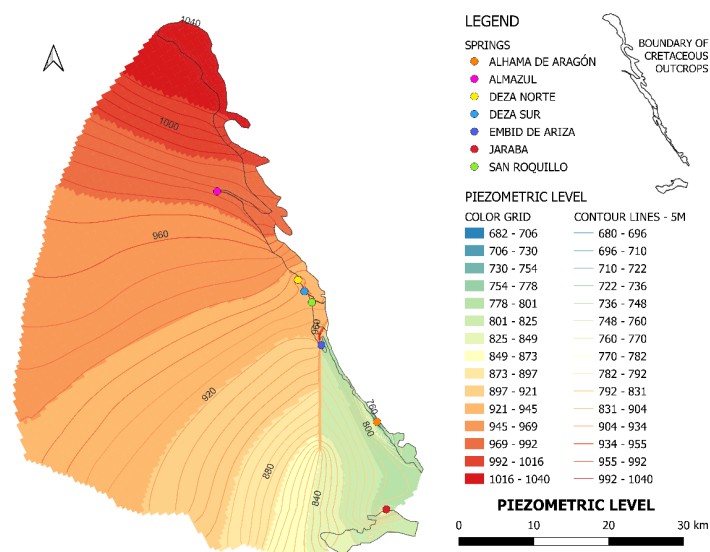

**Fig. 15.** *Result of piezometric levels after calibration*

Calibration elements used to adjust the mathematical model are shown in Fig. 16 and correspond to those previously indicated. Firstly, the piezometric levels of the observation points of the "Confederación Hidrográfica del Ebro" and secondly, the flows of the springs.

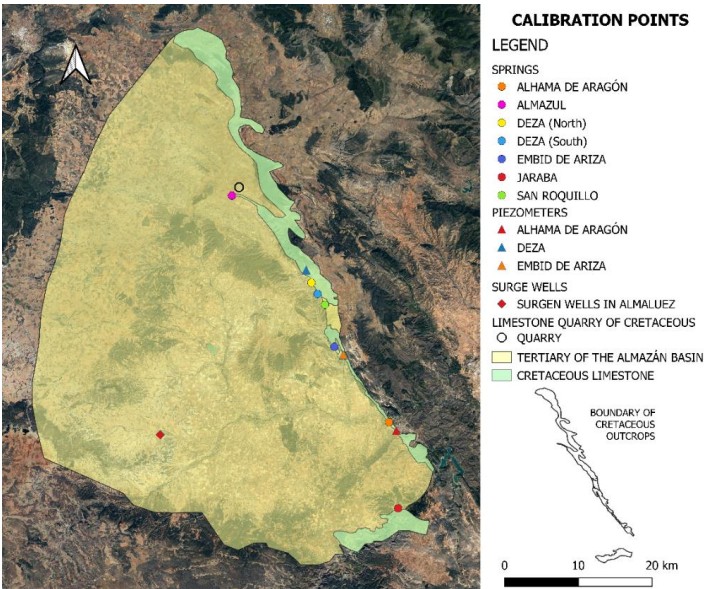

**Fig. 16.** *Piezometric calibration elements used in the modelling of the hydrothermal system (© Google Earth.).*





*4.3.7    Piezometric calibration.*

Calibration consisted of carrying out a series of simulations in which the model parameters (recharge, permeability, etc.) indicated in the conceptual model were varied in order to adjust the simulation results to the values established at the observation points, reducing the residual error between the simulated and observed values as much as possible.

Observation points have been obtained from the piezometers of the Confederación Hidrográfica del Ebro (CHE, 2023). These piezometers, as shown in Fig. 14, are located in Alhama de Aragón, Embid de Ariza and Deza. According to the data of the Confederación Hidrográfica del Ebro, the Alhama de Aragón piezometer is located at an elevation of 751.2 masl and presents a historical record of the variation of the piezometric level that fluctuates between .665.49 and 667.54 masl. The Embid de Ariza piezometer is located at an

elevation of 780 masl and the measured piezometric level varies between 777 and 773 masl. The Deza piezometer is located at an altitude of 1000.3 masl and the piezometric level measured varies between 919 and 925 masl.

The levels measured at the observation points correspond to the following, all measured in units of meters above sea level (m.a.s.l.):

**Table 2.** *Calculated vs observed groundwater heads at calibration points.*

| Alhama de Aragón piezometer | | Embid de Ariza piezometer | | Deza piezometer | |
|---|---|---|---|---|---|
| Calculated values | Observed values | Calculated values | Observed values | Calculated values | Observed values |
| 670 | 665 – 667.5 | 791 | 777-773 | 920 | 919 – 925 |

As can be seen, the calibration of the observation points is considered acceptable, as the measured and simulated values are very close (Table 2 and Fig. 17.).



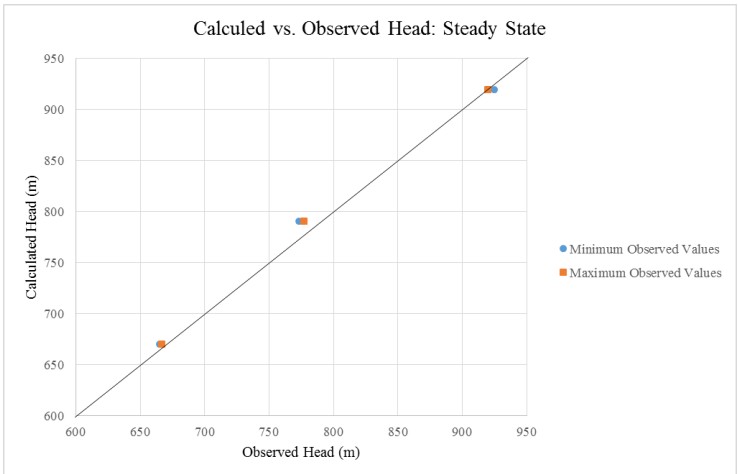

**Fig. 17.** *Graph of the calculated vs observed groundwater heads at calibration points*

It should be pointed out that there are very few observation points available, so other indicative calibration elements have been used to help with calibration. For example, in the village of Almaluez there are 3 surgen wells, and although no pressure data are available, they have served as a semi-quantitative calibration point. In fact, care has been taken to ensure that the piezometric Isopieces in the calibrated model are always above the topographic surface. As a counterexample, there is a Cretaceous limestone quarry near the village of Almazúl between the Tertiary, which is a window to the aquifer at 1050 m and through which water never emerges. Thus, this point has also served to ensure that the Isopieces of the model could never be above the bottom of the quarry.

### 4.3.8    Calibration of spring flow rates.

As indicated above, the information on observation points is scarce, and to guarantee a better and complete fit of the model, the flow rates of the different thermal springs have been used as a calibration tool. It should be pointed out that the values of the flow rates measured for the springs shown in Table 3 have a certain degree of uncertainty, since, although they are springs that are extraordinarily constant over time, it is not easy to measure them in the field. However, these are the ones that appear officially and in publications (ITGE-DGA, 1994).

As with the observation points, the objective is to adjust the simulation results to the real flow values observed in the springs. In this way, the calibration of the model is completed by adjusting both the simulated flow and piezometric values, reducing as much as possible the residual error between the simulated and observed values. The springs with which the model has been calibrated are shown in Fig. 16 and Fig.16.

**Table 3.** *Comparison of gauged and simulated springs in the calibration process.*

| Springs | Gauged flows (l/s) | | Simulated flow rates (l/s) | Deviation |
|---|---|---|---|---|
| | Maximum | Minimum | | |
| Jaraba | 500 | 600 | 609.81 | 2% |
| Alhama de Aragón | 434 | 520 | 551.85 | 6% |
| Embid de Ariza | 20 | 40 | 42.5 | 6% |
| Deza (South) | 120 | 140 | 102.13 | 0% |
| Deza (North) | | | 29.28 | |
| San Roquillo | 5 | 10 | 10 | 0% |
| Almazul | 5 | 10 | 6.37 | 0% |

As can be seen, the calibration of the flow rates is more than acceptable, presenting a really good fit (Table 3 and Fig. 18.). Therefore, according to the piezometric and spring flow calibration, the mathematical model can be considered adjusted.

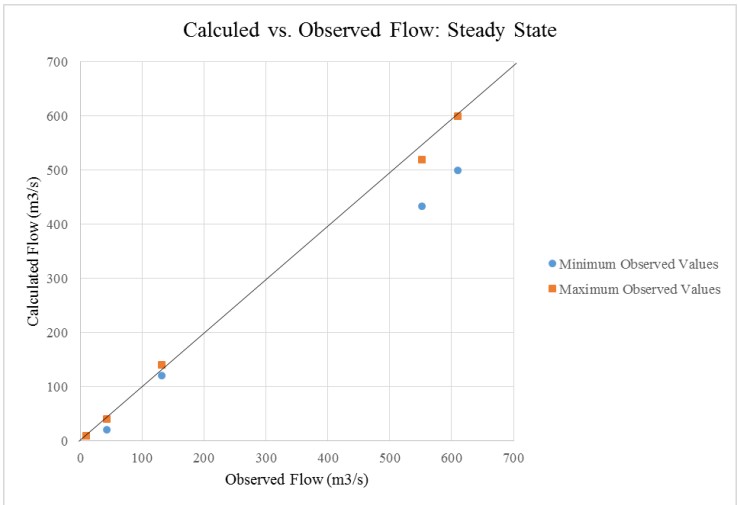

**Fig. 18.** *Graph of the calculated vs observed groundwater flows at Springs*

*4.3.9    Transit time estimation.*

Thanks to mathematical modelling of the hydrothermal aquifer, it has been possible to approximate the effective porosity of the hydrothermal system as well as the Darcy velocity.

In order to carry out this simulation, the Modpath package was used to simulate the path of a contaminant plume along the aquifer. This has allowed us to estimate the time it takes for a pollutant to travel through the aquifer until it emerges at the discharge points. For this purpose, a pollutant located in the northernmost area of the calcareous outcrops of the Aragonese Branch has been defined.





It has been observed that for a porosity of 0.3% the water takes approximately one century to reach the thermal springs of Alhama de Aragón and Jaraba. In the case of Deza, a duration of 20 years has been estimated, which coincides perfectly with the time estimated by tritium, as can be seen in section 4.4.

### 4.4    Hydrochemistry

#### 4.4.1    Isotope Hydrochemistry

#### 4.4.1.1    Tritium hydrochemistry

1.    Available data and tritium sampling

Given the fractional discharge of the thermal system, through several springs that are distributed along the Cretaceous contact of the Aragonese Branch with the Tertiary of the Almazán Basin, the tritium values of these springs have been used to estimate the age of the groundwater and its chronological distribution in

space. The evolution in time of the tritium content of the waters of the springs of the aquifer has also been studied. For this purpose, all the data on tritium values of the springs over the last 40 years (from 1981 to 2021) have been compiled and are shown in Table 4.

Regarding the origin of the data and sampling, the following should be pointed out: 1. As can be seen in the

above table, the values do not correspond to a continuous and systematic sampling, since they come from studies carried out by different organizations and years, but they do cover almost all the springs or groups of springs over time. Indeed, the 37 data in Table 4 come from our own investigations (year 1992, Sanz and Yélamos, 2000) and year 2021 (for this work), as well as from reports of the IGME and ITGE-DGA (1994) and other unpublished reports on tritium content obtained from the Pallarés Hot springs in 1982. 2. In the

most important groups of springs of Jaraba and Alhama de Aragón, many discharges were initially analysed; afterwards, sampling continued with those springs where there was certainty that they were of deep flows and not mixed.3. Going down to the concrete, it has been observed that in Embid de Ariza (Table 4, year 1992) the most representative samples are those obtained from the catchment gallery and not from the washing place 500 m downstream and that the water arrives through a ditch where there is a contamination

with modern sub-valve water. 4. In the springs of San Roquillo, the representative sampling of the regional flows should be done during low water levels, since it has been observed that there is mixing by recent water infiltrated during rainy months.

2.    Groundwater age.

For the various estimates of the age of the water in the springs, it is assumed that the Alhama aquifer

approximately follows a piston flow model. As can be seen from the results of the conceptual model and the flow modelling, recharge is assumed to occur preferentially in the north-western sector of the Cretaceous calcareous aquifer, where the groundwater moves without major mixing processes with the Tertiary. The tritium content of the groundwater is therefore mainly controlled by the laws of radioactive decay.

The estimated age of the water of the Alhama and Embid springs (1993) for the years 1981/1982 and 1993

according to the ITGE-DGA, 1994) is that they corresponded to recharge waters prior to 1952. In 1982 and based on 12 monthly analyses throughout the year in the spring of the Pallares Hot Springs gallery, the tritium





content was practically nil; it can therefore be assured that the Alhama springs were not influenced by water from after 1952, nor by recharge from nearby outcrops.

Most of the Jaraba springs, at least the warmest ones, also correspond to recharges prior to 1952, although there are others with values that reflect mixing with more recent water. This mixture of waters is characteristic of Jaraba: springs with a temperature range between 18ºC and 32ºC, tritium isotopic variations, greater dispersion in the composition of the waters.

This is logical, since the non-thermal Cretaceous limestone aquifer, which is here folded and sub-horizontal and expands to the southwest over an area of about 27 km$^2$ in the vicinity, drains cold water (between 12º and 875    14ºC) also at the lowest point of Jaraba. The Mesa River runs in a gorge parallel to the termination of the deep Cretaceous in contact with the Tertiary for about 2 km, so the interaction of cold water from the river and discharges is important. This is not the case with the river Jalón in Alhama de Aragón, which cuts perpendicularly to the vertical layers of Cretaceous limestone in a minimum stretch of no more than 200 m, and there is no option for there to be much mixing of water. The ages obtained in the 1992 sampling for the 880    Deza and San Roquillo springs show that the waters possibly infiltrated in the 1980s (Sanz and Yelamos, 1998). For the 2000 sampling, the age of the waters of Deza and San Roquillo infiltrated about 12 years ago, and in Embid and Alhama, 36-40 years ago; values similar to those obtained by Sanz and Yelamos (1998).

For the last sampling in the year 2021, as shown in Fig. 19, where the tritium activity values have been represented, together with the values of tritium activity in precipitation in the station closest to the sampling 885    points, Zaragoza, belonging to the Spanish Network of Isotope Monitoring in Precipitation (REVIP), managed by CEDEX and AEMET, and also the one in Madrid, used as a reference. In Madrid there are monthly measurements of tritium activity in precipitation from 1970 to 2020, with some information gaps, while in Zaragoza there are only annual weighted measurements from 2000 to 2019. In both cases the annual mean weighted with the amount of precipitation has been plotted. The values before 1970 and the gaps have 890    been completed for the Madrid station with the model of tritium in precipitation in the northern hemisphere according to the criteria of Plata (1994, 2006) and Diaz et. al, (2009).

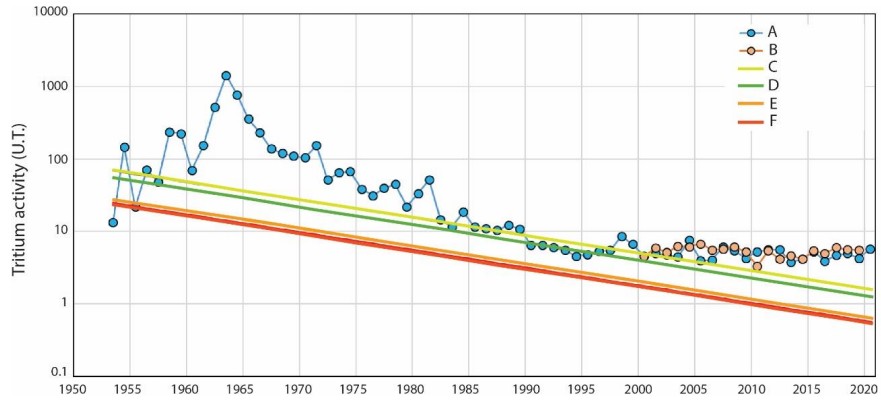

**Fig. 19.** *Comparison of tritium values of groundwater samples with those of annual precipitation in Zaragoza and Madrid, belonging to the REVIP period 1953-2018 in semi-logarithmic scale. (A.* 895    *Precipitation in Madrid. B. Precipitation in Zaragoza. C. Deza springs. D. San Roquillo springs. E. Springs of Alhama de Aragon. F. Jaraba spring).*





The straight lines in Fig. 19. link the tritium activity values of all recharge waters that could give rise to the measured value in each sample after radioactive decay. Any cut-off points of these decay lines with those linking the tritium activity values in the precipitation water have the same probability of corresponding to

these recharge waters, depending on the flux model. In this case, and considering a piston flow model, both the waters of Embid and those of the Alhama de Aragón springs would correspond to waters recharged prior to the 1960s, i.e., more than 61 years old. Those of the San Roquillo and Deza springs would come from a recharge in the early 1990s or, at least, before the year 2000, i.e., some 20-25 years old.

These results from the 2021 campaign are somewhat older than the age values obtained from previous

campaigns but remain in a similar range. In fact, it is thought that the ages of Alhama and Jaraba are close to a century according to simple analytical calculations of the real velocity according to the length of the flow lines obtained from the groundwater contour map of the mathematical model (Fig. 15), the hydraulic conductivity and the porosity obtained from the modelling (0.3%). It is important to note how the simulated ground water contour could be assimilated to isochrones and fit very well to the progressively increasing age

model Deza-San Roquillo-Embid de Ariza-Alhama/Jaraba, according to the regional flow.

**Table 4.** *Tritium isotope values (in TU).*

| Springs | 1981 (1) | 1982 (2) | 1992 (3) | 1993 (1) | 2000 (4) | 2021 (5) |
|---|---|---|---|---|---|---|
| Deza springs | | | | | | |
| Suso spring | | | 7,2+0.13 | | 4,12+0.12 | 1,57+0.29 |
| | | | 8,1+1.0 | | | |
| El hocino spring | | | 8,0+0.13 | | | |
| | | | 7,4+1 | | | |
| San Roquillo | | | | | | |
| San Roquillo spring | | | 1,23+0.5 | | 3,72+0.19 | 1,25+0.28 |
| Embid de Ariza | | | | | | |
| Embid de Ariza gallery | | | | 0,74+1.37 | | |
| Lavadero | 2,9+1.2 | | 3,3+0.9 | | 1,99+0.16 | 0,57+0.27 |
| | | | 3,2+0.11 | | | |
| Alhama springs | | | | | | |
| El chorrillo (termas san Roque) | | | | 5,8+1.6 | 1,28+0.15 | 0,63+0.28 |
| San roque (B. Cantarero) | | | | 6,0+2.0 | | |
| Gallery (termas Pallares) | 0,6+1 (1) | | | 6,7+1.4 | | |
| Los baños spring (B. Guajardo) | | | | 3,2+2.0 | | |
| Termal lack | 0,2+1 | | | 5,9+1.6 | | |
| Baños del rey | | | | 8,4+1.6 | | |
| Jaraba springs | | | | | | |
| Virgen de las Nieves spring | | | | 10,7+1.4 | | |





| Springs | 1981 (1) | 1982 (2) | 1992 (3) | 1993 (1) | 2000 (4) | 2021 (5) |
|---|---|---|---|---|---|---|
| San Vicente spring (B. Sicilia) | | 0,0+1.1 | | 5,4+1.8 | | |
| San Luis spring (B. de Serón) | | 0,0+2.3 | | 1,9+2.0 | | |
| San Antonio spring | | 0,0+1.1 | | 3,3+1.4 | | |
| San Roque/Sta Dorotea (b. de seron) | | 0,0+1 | | 3,8+2.0 | | |
| El Prado spring (B. de Serón) | | 0,0+2.4 | | 10,0+1.4 | | |
| Sondeo "Cañar 2" | | | | 6,6+1.4 | | |

(1) No tritium content was observed during the 12 months of 1981 (proc.: Pallarés Hot springs).

(2) Taken from ITGE-DGA (1994)

(3) Taken from Yélamos and Sanz (1998)

(4) This work

3. Evolution of tritium content over time

On the other hand, the tritium content of some of the springs under study has been monitored over a long period of time and is shown in Fig. 18. The following observations can be made:

- The shape of the tritium content curve of all the springs is a smoothing of the rainfall curve and
shifted in time. In all cases, the upward curve is steeper than the downward curve, with a gentler slope, and similar in shape to that of the rainfall. We can see how the decreasing trend is maintained with a similar slope in all cases, i.e., they are sub-parallel.

- Curves values are higher in the springs with more modern waters and progressively decrease in those with older waters (Deza/San Roquillo-Embid-Alhama/Jaraba) according to the underground
flow, but in all cases, there is a certain mixture of post-1952 waters from 1982 onwards. Indeed, it can be seen that, in 1982, the thermal springs of Alhama and Jaraba were not influenced by water from after 1952, nor by the recharge of nearby outcrops, since their values are below 0.5 TU. It is from 1985 onwards when the mixture of water infiltrated after 1952 begins to be noticed, but the values are still very low. In Embid it would probably be a little earlier, around 1979, if a curve with the same function as that of Jaraba and Alhama is drawn. In Deza and San Roquillo, it would also
be earlier, around 1975, using the same procedure. There is a maximum tritium content in the Embid, Jaraba and Alhama springs around 1992, displaced with respect to the maximum tritium content of the rains by 32-33 years. All this leads us to think that we are dealing with the same aquifer system whose recharge is far away, which has a large reservoir of old water, and that the recharge of young
water has not been mixed to any great extent with the former because it is drained by the Deza and San Roquillo springs, purifying the system and causing long, deep and old flows to reach Alhama.

- Thermal springs of Jaraba and Alhama de Aragón share the same or similar temperature, chemical composition, and geological contact of emergence. Their tritium isotopic composition and their evolution over time are practically the same, as shown in Fig. 20. It's D and $O^{18}$ isotopic composition
is also very similar. All this shows that there is a hydrogeological parallelism and the same origin





between both springs: as can be seen in the ground water contour map (Fig. 15), both springs share the same recharge area of similar altitude and constitute the end of flow tubes of similar length and flow rate.

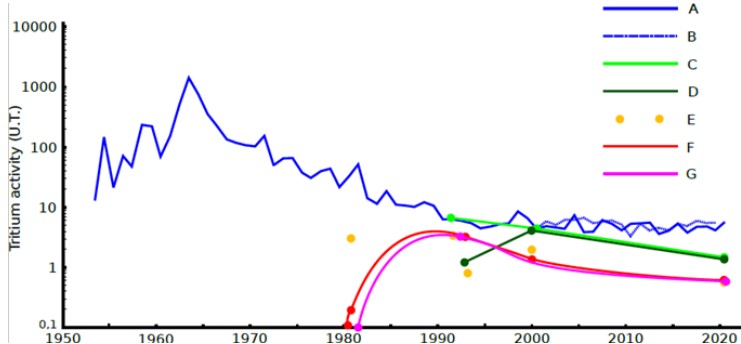

**Fig. 20.** *Evolution of the tritium content of precipitation in Madrid and in the springs of the Alhama de Aragón thermal aquifer (A. Precipitation in Madrid. B. Precipitation in Zaragoza. C. Deza springs. D. San Roquillo springs. E. Embid de Ariza. F. Springs of Alhama de Aragon. G. Jaraba spring (own data and from IGME 1985, Sanz and Yelamos, 1998, ITGE-DGA, 1994).*

*4.4.1.2    Information provided by Deuterium and Oxygen [18].*

Table 5 presents the collected values of D and $O^{18}$ and the new values for the year 2000 made for this work. We consider the data obtained to be valid since, for hydrothermal systems with low to medium temperatures, as is the case, the values of $\delta D$ and $\delta 18O$ do not undergo substantial changes, which does not occur in those with high temperatures (Andre et al, 2005). It can be seen that they do not differ substantially from the previous ones, which confirms that the recharge area is located in the same zone for all the springs. According

to the rainfall data from a relatively close station in Madrid, which corresponds to recharge areas located between altitudes 1000-1300 m. Since both the Sierra del Solorio and the area of the Cretaceous outcrops in the Aragonese branch coincide with these altitudes, this isotope does not serve to discern which of the two areas is truly the recharge area. It is significant how the chloride content increases with flow, while $^{18}O$ remains almost unchanged.



**Table 5.** *Environmental isotope values of D and O18 for thermal and semi-thermal springs*

| Springs | 1985 (1) | | 1992 (2) | | 1993 (2) | 2000 |
|---|---|---|---|---|---|---|
| | $\delta^{18}O\ ^0/_{00}$ | $\delta^2H$ | $\delta^{18}O\ ^0/_{00}$ | | $\delta^{18}O\ ^0/_{00}$ | $\delta^{18}O\ ^0/_{00}$ |
| | | | Spring | Summer | Spring | |
| M$^{al}$ de Alhama (Termas Pallarés) | | -62 | -8,71 | -8,74 | | -8,68 |
| M$^{al}$ de Alhama de A. (El Lago) | | -60 | | | | |
| Embid de Ariza (galería) | | -59 | -8,74 | -8,73 | -8,46 | -8,58 |
| San Roquillo | | | -8,91 | -9,03 | -8,48 | -8,73 |
| M$^{al}$ de Suso (Deza)º | | | -9,03 | -9,02 | -8,54 | -8,86 |
| M$^{al}$ de Ocino (Deza) | | | -8,94 | | | |

(1)  From IGME /1985)

(2)  From Yélamos y Sanz (1985)

*4.5    Discussion about the origin of the springs*

As mentioned in the Introduction, the hydrogeological conceptual model of the Alhama-Jaraba thermal system was not yet clearly established. Initially it was thought that the main flow of thermal waters came from the south (Sierra de Solorio) and not from the north and northwest. To the south there is a high plateau with extensive outcrops of carbonate materials, so the recharge water would flow through the Upper Cretaceous calcareous aquifer, which lies below the low-permeability Tertiary materials of the Almazán

basin, which would serve as aquitard, and then rise rapidly to the Deza, Embid, Alhama and Jaraba springs from great depth along the Cretaceous alignment, taking advantage of the vertical arrangement of the limestone layers (Sánchez et al. , 2000). Analysing the results of the present study it can be concluded:

1. Recharge in the Sierra del Solorio is overwhelmingly from Jurassic rather than Cretaceous carbonate
outcrops, which is where all the thermal and semi-thermal springs are located. More than 95 % of the recharge area of the Solorio Mountain is formed by outcrops of Jurassic carbonate materials, and the rest by carbonates of the Upper Cretaceous aquifer and which are on the border with the Tertiary of the Almazán Basin.

2. The Jurassic carbonate aquifer is conserved almost in a natural regime and its hydraulic balance has
been established quite clearly in previous works (SGOP, 1990);De Toledo and Arqued, 1990), so there is no surplus water to justify the large thermal flows of Alhama, Jaraba and the rest of the springs, which add up to some 1,200 l/s.

3. Even assuming large errors in the calculation of the balance of the Solorio Mountain from these earlier studies, it must be admitted that the water recharged mainly in the Jurassic should somehow





be transferred to the Upper Cretaceous limestones above. But as seen above, the Jurassic does not exist beneath most of the hydrothermal aquifer (Figure 2) except for a narrow strip of 5 km north of the border of the Castellana Branch with the Tertiary, disappearing from Jaraba onwards. This aquifer is therefore quite far from the springs of Alhama, Embid, San Roquillo and Deza, making such a transfer practically impossible. And even if it could be done in the aforementioned strip, it

would first have to cross some 200 m of poorly permeable lithology separating the two groups (sands and marls from the base of the Cretaceous). To solve this question, one would then have to resort to the supposed hydraulic connection between the two aquifers, which could be provided by hypothetical fault jumps, and/or to the flow mostly concentrated through these faults (Hernández Pacheco, 1954, ITGE-DGA, 1994). In Hernández Pacheco's work, great importance is given to the

circulation through faults as the origin of the springs, such as the Alhama fault(s), but in the simulations carried out specifically in this mathematical model, the faults do not justify large circulation flows.

     4.   It is simpler and more logical for the water to be recharged in the Cretaceous aquifer itself, especially in the Rituerto basin, where it was not known until now where the recharged water went, nor had

any hydraulic balance been established. And that it then circulated through the deep zone or hot spot on the border of the Aragonese branch with the Almazán Basin and emerged without changing aquifer. Note that the thermal and semi-thermal springs are all on the border with the Aragonese branch, not in the Castilian, branch which is the deep, hot zone.

     5.   If the flow was from the Sierra del Solorio towards Jaraba and Alhama, it would be logical that the

flow would be concentrated at the lowest point (Alhama) and not in 5 springs that are 30-40 km apart. Furthermore, there is no hydraulic gradient that would justify the water reaching areas so far away from Deza or San Roquillo (elevations of approximately 900 m), as the underground discharges in the Sierra del Solorio are also located at around 900 m.

     6.   The higher temperatures of the Jaraba and Alhama springs are not justified, as the flow does not

pass through the deeper areas of the aquifer. It would be the other way round, as for the flow to reach Deza and the rest of the springs, the flow lines would have to pass through deeper areas. The deep wells that reach the Cretaceous on the edge of the Castilian branch are not thermal water wells, unlike some located in the Aragonese branch.

     7.   Nor are the values of groundwater age obtained from tritium for Alhama/Jaraba justified because

the flow distances from the centre of gravity of the Sierra del Solorio to these springs (about 25 km) are not so large as to result in ages above 60 years, as occurs in our model with flow lines of 70 km. Neither does it explain the order of increasing age of Deza-San Roquillo-Embid- Alhama/Jaraba, it would have to be the opposite, as the length of the flow lines increases towards Deza. Furthermore, if the flow first passed through Jaraba and ended in Alhama (ITGE-DGA, 1994), the age of the latter

spring would be greater, and yet its age and evolution is totally parallel to that of Jaraba.

## 5   Conclusions

Knowing the origin of springs in underexploited aquifers can be a difficult problem to solve due to the uncertainty caused by the lack of sufficient hydrogeological data. Defining the feeding zone of an aquifer is





the first step to evaluate natural recharge and the protection zone against contamination and unsustainable
exploitation of groundwater. This is the case at hand, a thermal aquifer in a natural regime that supports the
most important spa complex in Spain, but whose origin was not known until now. The aquifer currently
presents very small degrees of exploitation and it is urgent to make these results known in order to apply
sustainable management in the future, with preventive conservation measures for both the quality and
quantity of groundwater.

The modelling of the regional flow of the thermal system studied has served to confirm and quantify various
aspects of the initial hydrogeological conceptual model and to know the origin of the most abundant thermal
springs in Spain. This has been possible thanks to the previous integration of diverse hydrogeological
information already existing or generated for this study: geometry of the aquifer in depth and application of
systems of representation by dimensioning, stratigraphy and absence of the Jurassic carbonate series that
limited the thermal aquifer to the Cretaceous, location of the heat source, origin of the porosity by
karstification of the Cretaceous limestones, age data of the waters according to content in isotopes of tritium,
etc. Among other aspects, the calibration of the model confirmed the low permeability of the Tertiary, and
also confirmed the recharge zone, which was under discussion, having quantified it. But it has also required
a careful review of the lack of hydraulic continuity in certain sectors of the aquifer due to fault jumps, and
the little importance of the existing faults as transport routes to explain by themselves the large volume of
water evacuated.

In the end, it is believed that this is a good example of the functioning of a low-temperature thermal system
in a natural regime associated with a calcareous aquifer that occupies the divide of two neighbouring basins
allowing subway hydraulic communication between them, and which can be explained thanks to the water
flow of the springs located in the basin at the lower altitude. The thermalism of the waters is explained by the
normal geothermal gradient due to the depths of up to 4,500 m reached by the flows in the aquifer. These
flows discharge progressively and partially through semi-thermal springs located on one of the edges of the
aquifer at progressively lower altitudes (Deza, San Roquillo, Embid), where the temperature and age of the
water also progressively increase. Finally, most of the flow follows a south-easterly direction, where the
basement of the Almazán basin and the "synclinorium" that structures this aquifer end. Here the flow is
discharged into the most distant springs of Alhama and Jaraba, some 70 km from the common recharge zone.
These are the springs with the highest flow, the highest temperature, and the oldest water, probably more than
60 years.

This applied methodology can be extrapolated to other similar cases in order to know in a preliminary way
the origin and hydrogeological functioning of thermal aquifers little or no exploited at all. This will serve to
identify the recharge area and to propose early sustainable management of water resource conservation, such
as the appropriate design of perimeters for the protection of singular thermal springs.

**Competing interests.** The contact author has declared that none of the authors has any competing interests

**Acknowledgments**. Part of the isotope analyses have been financed thanks to grants from the Research
Groups of the Universidad Politécnica de Madrid (references: VAGI23ESP and VAGI20ESP).



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
