# Peer review of "Improvement of the thermal spring protection area through numerical modelling and interdisciplinary studies."

_Hydrology and Earth System Sciences, 2024_

## Referee Comment (RC2)

[referee-annotated manuscript omitted]

---

## Author Comment (AC1)

I read this article with pleasure. Regarding its content, 10 observations caught my attention.

Contents

1.- The reviewed article (*1) constitutes an original and unpublished investigation.

2.- The work focuses on the question of the hydrogeological functioning of a thermal aquifer system that is almost in a natural regime.

3.- The Alhama de Aragón aquifer is probably the most important in the Iberian Peninsula, at least for its hydraulic resources.

4.- The Alhama spring is one of the largest thermal springs in Europe and supports an important spa industry.

5.- On the other hand, the authors have compiled all the existing information.

6.- In addition, numerous previous studies on different topics are used (deep geophysics, detailed geology, etc.).

7.- The work has been complemented with field work that has lasted several years (inventory of water points, monitoring of isotopic hydrogeochemistry, etc.).

8.- All the information has been integrated and used to design the hydrogeological conceptual model.

9.- The flow model has been simulated with a numerical model.

10.- And finally, the results of the numerical model have confirmed the conceptual model and the origin of the springs.

(*1) "The origin of Alhama de Aragón and Jaraba thermal springs. Numerical modeling of the regional flow of the geothermal systems, Almazan Basin, Iberian range, Spain"

Personal assessment.

In my opinion, this is a work that is of interest to the international hydrogeological community.

11.- Perhaps the greatest interest of the work is the methodology.

The use of quality data and robust tools has made it possible to locate the recharge area.

12.- This work lays the foundations for the protection of important springs.

13.- Furthermore, the study is timely. The aquifer is not exploited. It is possible to apply sustainable management.

Recommendations

I recommend publishing it with some minor modifications. Some bugs need to be fixed.

14.- In the legend of figure 1, put "Pyrenees" instead of Pyrenees.

It has been corrected. You can check it in the final manuscript where your comment has already been taken into account.

15.- The toponymy mentioned in the text must be included in the figures. It will make the text easier to understand.

* Ibdes (line 107) It has been included in Figure 1. You can check it in the final manuscript where your comment has already been taken into account..

* Sierra del Solorio (line 120, 270, Figure 3) It has been included in Figure 1. You can check it in the final manuscript where your comment has already been taken into account.

* El Raido (line 211) It has been included in Figure 1. You can check it in the final manuscript where your comment has already been taken into account..

* Calmarza (lines 684, 704 and 710). I suggest including the toponym in figure 3, box D. It has been included in Figure 1. You can check it in the final manuscript where your comment has already been taken into account.

16.- Line 307, put the number 3 as super index (hm^3) It has been corrected.. In addition, a general review of the manuscript is made to check that there are no more typographical errors such as the one indicated. You can check it in the final manuscript where your comment has already been taken into account.

17.- Lines 358 and 360. Reference is made to (T). What does (T) mean? It's confusing. Typographic error. It is eliminated. You can check it in the final manuscript where your comment has already been taken into account.

18.- Table 4. You must include proper nouns in capital letters. Table 4 is updated including proper nouns in capital letters. You can check it in the final manuscript where your comment has already been taken into account.

19.- Hocino Springs does not have an H. Indeed, this error is corrected. You can check it in the final manuscript where your comment has already been taken into account.

20.- The word 'roof' is frequently used. It seems to me that the appropriate term would be 'overhead'. Confirm, please. We think "roof" is the appropriate term. We have consulted it in order to confirm it

21.- Figure 4-C. Indicate in the figure caption that it is a draft that shows the structural analysis carried out, using the dimensioning technique. Based on Referee #2's review, we have removed the entire figure.

Suggestions

The work would benefit if the information were briefly expanded in some aspects:

We appreciate these suggestions, although we are going to answer a little more extensively:

22.- The absence of Jurassic is a key observation to understand the origin of thermal springs. It would be appropriate to add some data to support this fact.

After the paragraph, on the line275 we could add:

"Toward the north, these gradually lose thickness, which is evident on the eastern edge, to such an extent that, in Jaraba, the thickness is minimal and has practically disappeared in Alhama de Aragón. From this locality, and all along the edge of the Aragonese branch up to the periclinal closure of the Cardejón anticline, the Jurassic is not present either. It is from Jaray onwards that they appear again and outcrop along the Rituerto polje as far as the Sierra de la Pica. Below the Tertiary of the Almazán Basin, and according to reflection seismic data and hydrocarbon exploration boreholes (Maestro, 2004), such as the Gredal borehole, the Jurassic does not exist in almost the entire Almazán Basin within the study area, except in the vicinity of the border with the Castilian branch. Between Arcos de Jalón and Jaraba, and according to the aforementioned reflection seismic, the Jurassic seems to have disappeared less than 5 km north of the contact between the Tertiary of the Almazán Basin and the Mesozoic of the Castilian branch. In hydrogeological exploration boreholes it has been detected about two kilometers south of Alconchel de Ariza and Cabalafuente (Zaragoza) (CHE, 2023)"

Could more details be given about the origin of the karstification associated with the Tertiary - Quaternary?

You can check it in the final manuscript where your comment has already been taken into account.

24.- Could you explain how the upward erosion of the Jalón River has contributed to the capture of the thermal aquifer and its underground transfer to the Ebro Basin?

After the the line 435, the following could be added to the article:

"Referring to the karstification of the Cretaceous limestones during the Neogene-Quaternary, and according to Rodríguez García (2008), after the elaboration of the Intra-Miocene Erosion Surface, corrosion of the created planes occurs, the formation of small cavities and deposits of terra rossa during the middle-upper Miocene. Subsequently, the sedimentation of lacustrine carbonates occurs in more humid conditions, probably thanks to the greater intensity of the dissolution

processes on the Mesozoic mountain edges. The upper Miocene-Ruscinian would be characterized by a decrease in the base level due to the exorheic opening of the basin and the action of tensional tectonics. This entails a stage of generation of poljes and corrosion surfaces, as would be the case of the poljes of Araviana (Sanz, 1987), Cañada Hermosa (Echeverría, 1989), Rituerto and Noviercas, in the upper Rituerto basin to the north (Sancho Ruiz, 2019). With the Ibero-Manchegan tectonic phases, during the Pliocene and favored by the associated fracturing, the main stage of poljes formation takes place in the southern mountains, such as the Layna poljes (Gracia et al., 1996).

Regarding the geometry and deep structure of the Almazán basin, it must be said that on this northern edge of the Aragonese Branch at the contact with the Almazán Basin, the structure has a general NW-SE orientation, unlike the edge with the Castilian branch, which is dominated by a W-E direction. At depth and based on the ground water contour of the Tertiary base obtained from reflection seismic surveys, oil drilling, and the support of gravimetric studies, the geometry at depth of the Almazán Basin, and specifically of the Cretaceous calcareous aquifer, can be deduced (Maestro, 2004). The structure of the basin is characterized by folds with NW-SE to E-W directions, associated in some cases with north-vergence thrusts. Two sectors are considered to the west of the Almazán-Soria meridian, the fold axes are E-W, while to the east, as they approach the Aragonese branch, they acquire NW-SE directions typical of this unit of the Iberian Range. In the eastern sector of this basin, three areas with different structures can be distinguished:"

You can check it in the final manuscript where your comment has already been taken into account.

25.- Figures 15 and 16. Why are two springs shown in Deza? It is necessary to clarify this duplicity.

Deza has a series of springs along the outcrops of the Cretaceous limestones, which correspond to the lowest points. In the model, 2 "DRAIN" boundary conditions have been modeled since there are two different areas of springs relatively close.

26.- Figures 17 and 18. It is advisable to put different symbols for each spring and include them in the legend. The figures are confusing.

With the intention of improving the quality of the figures, we have improved them significantly, including all the advice you have provided us. It is appreciated. You can check it in the final manuscript where your comment has already been taken into account.

I show you below how the updated figures have turned out.

[Figure]

[Figure]

.27.- Line 814. The text 'fig16' is repeated. It is a typographic error. It should we said "Table 2 and Fig.15). In addition, a general review of the manuscript is made to check that there are no more typographical errors such as the one indicated. You can check it in the final manuscript where your comment has already been taken into account.

28.- Almost all figures include text with a very small font size. It is difficult to read.

A general review of the Figures of the manuscript is made to check it.

---

## Author Comment (AC2)

**REPLY TO THE COMMENTS OF THE REFEREE #2**

I appreciate the timing and efforts of the authors in the preparation of this manuscript. There is so much geological and hydrogeological information provided in the paper, and in my opinion, all of them are valuable pieces of information. Overall, the manuscript needs significant improvements, particularly in the writing format. The submitted paper is prepared like a thesis, the number of sub-sections is too much, which makes the paper unreadable. Some of these sub-sections should be merged appropriately. The authors must re-organize the whole paper according to the manuscript format.

**Thank you very much for the time and effort spent reviewing this article. We believe that we have taken into account each and every one of the observations made to us.**

**As recommended, an effort has been made to unify subsections and unnecessary information has been removed where possible.**

Secondly, this paper is mainly focused on the improvement of the thermal spring protection area through numerical modeling and interdisciplinary studies, however, when I was reading the manuscript, I felt like reading a regional study, pointing out the importance of a local geothermal system. The paper needs to address what are the new methods to better reveal the protection areas by comparing the existing methodology and approaches. What are interdisciplinary studies currently available (e.g. hydrochemistry and environmental isotopes), or newly used? Are there similar applications in the literature? The novelty of the study (if available) should be emphasized. In my opinion, the authors focused on the modeling phase too much, which shaded other sections.

**As a preliminary phase to delimiting the protection areas, the Introduction briefly addresses what interdisciplinary methods exist to identify the recharge areas of springs, and what this article is about. These methods are referred to in our particular case, partly to show that they are not overshadowed by the mathematical model. Classic references and relatively recent examples that have been recommended are cited. And the novelty of the present study is emphasized, which constitutes an example where not only have almost all classical techniques been applied, but other innovative ones have been integrated, such as the existence of widespread paleokarstifications in the thermal aquifer and a mathematical flow model.**

Considering these major comments and minor recommendations (added in the pdf file), I recommend a major revision.

Kind regards,
The reviewer.
 Major comments

**With the intention of facilitating the review process, we have worked hard since receiving the reviewers' comments to update the manuscript with the comments you have sent us.**

**In any case, we respond one by one to the comments that have been made to us, and we greatly appreciate them.**

The introduction must be tidied up. State of the art is not shared, and the aim of the study is not clear.

"As said before, the introduction briefly develops the state of the art on interdisciplinary methods to delimit the recharge areas of springs, as a preliminary phase to the protection areas. Classic references and relatively recent examples that have been recommended are cited.

The objectives are clarified, highlighting the main objective.

Since we cannot attach the manuscript, we paste here part of the introduction that we have updated.  The following paragraph will be located after line 65.

"Thus, the first question that must be resolved for the protection zoning of an aquifer system that drains a spring is to find out its catchment area or recharge area. The delimitation of catchment areas requires knowledge of a combination of topographical, geological factors, hydrogeological and hydrological considerations, etc. The geometry of an aquifer is generally defined by stratigraphic, tectonic and topographic elements. Indeed, an aquifer that drains towards a spring is a geological body, characterized by its geometry and internal structure, hydrogeological properties and hydraulic boundary conditions

To understand these characteristics, a variety of research methods can be used, such as hydraulic balances, natural tracers and artificial tracer tests; In karst hydrogeological systems, allogeneic basins should also be taken into account. (Goldscheider and Drew, 2007, Goldscheider, 2010).

Topography has an important influence on groundwater flow patterns and the recharge area of a spring is always situated above the spring level, although flow can also occur below. This level, with flow lines rising from deeper areas of the aquifer towards spring discharges (except in volcanic areas) are the most obvious manifestation of hot groundwater rising to the surface, although the recharge area is always It is located at a higher altitude; We will see this in our case. We will also see in our karst system how the relationship between topography and the dividing lines between neighboring hydrographic basins is not so simple, since the flow of groundwater from the thermal aquifer crosses below the Duero-Ebro dividing line.

Determining the water balance helps quantify the size of the spring catchment area. In general, a water balance not only provides the minimum size of the spring catchment area, but also the transfer from other aquifers or the upward flow of deep groundwater in large-scale regional flow systems. All of this can further complicate or limit the application of water balances, but in our case we have validated it with the application of a flow model.

Natural and artificial tracers can help delineate the spring catchment. Natural tracers include water temperature, which has been underutilized but may have great potential (e.g., Anderson, 2005, discussing it for large sedimentary basins; Wagner et al. (2014) in field experiments). . As will be seen, in our case the correspondence of the depths reached of the flow lines of the system with the different temperatures of the springs associated with each of them is very clear. The chemical composition of water, stable isotopes, and other parameters are also included as natural tracers (for example, and for a low-temperature thermal system: Pasvanoğlu and Çelik (2019). The stable isotopes of the water molecule

(18O and D) are often used to determine the average altitude of the recharge zone, especially useful in mountainous regions. Tritium (3H) can also inform us about the age of groundwater and therefore about the distances necessary to explain them, that is. , the location of the charging area.

The existence of a geothermal system requires, in addition to a heat source, the presence of permeable geological units that form aquifers or reservoirs, as well as adequate water recharge to the system. In this sense, it is essential to know and study the hydrogeological aspects in these geothermal systems (e.g., Cappacionni, et al., 2011, Chandrajit et al., 2013 for isotopic and other techniques). Or in Szocsa et al., 2018, where a multidisciplinary study is also carried out to investigate one of the most important thermal aquifers in Europe.

In this case, the most innovative aspect of our work is that most of the techniques mentioned above have been applied in an integrated manner, from the study of paleokarstification (which justifies the existence of the thermal system as an aquifer), the definition of the geometry of the aquifer through geophysical studies and application of bounded plans, etc. The assimilation of the results of all these interdisciplinary methods have not contradicted each other, but, on the contrary, have allowed the definition of a new conceptual model that has been contrasted by a mathematical groundwater flow model. With this, a new detailed location of the protection areas of the thermal springs of Alhama de Aragon and Jaraba has been achieved."

The majority of the given information in the introduction is related to the site description and should be moved to the Study Area section.

La mayor parte de la información proporcionada en la introducción está relacionada con la descripción del sitio y debe trasladarse a la sección Área de estudio.

This is done, it is moved to the Study Area. . You can check it in the final manuscript where your comment has already been taken into account.

The study area section is too long to read and understand. There is too much (unnecessary) information and details are shared. There are lots of sub-sections, and I recommend merging them into the "Geology" and "Hydrogeology" sub-sections.

Subsections are merged, leaving only Geology and Hydrogeology. It has been shortened somewhat, but it is not easy. . You can check it in the final manuscript where your comment has already been taken into account.

For instance:

"4.3 Simulation of groundwater flow in the thermal aquifer" is enough for a header. Do not divide these sections into sub-pieces (4.3.1/ 4.3.2/ 4.3.3…) Give all the necessary information by summarising. This is what I mean:

4.3.4 Hydrogeological parameters are given in separate sub-sections (4.3.4.1. or 4.3.4.2….) These details do not make the paper better, please decrease the resolution of the details in the paper. Please merge these sub-sections as much as possible.

**Subsections are merged as recommended, and summarized to the extent possible. . You can check it in the final manuscript where your comment has already been taken into account.**

The results and discussion section includes too much information which should be given in "Model Setup".

**We agree with your comment. And we move part to chapter 3.2 as indicated. . You can check it in the final manuscript where your comment has already been taken into account.**

Figure 3 is a very well-prepared hydrogeological map, however, Figure 1 and Figure 2 should be merged into 1 figure. Figure 4 (in my opinion) is not necessary and could be removed from the manuscript.

**Figures 1 and 2 are merged. Figure 4 is removed as recommended. You can check it in the final manuscript where your comment has already been taken into account.**

The sub-section "4.3 Simulation of groundwater flow in the thermal aquifer" should be given in the "3.2 Modelling of hydrothermal system flow" sub-section. Please describe the model before giving the results.

**Subsection 4.3 is included in subsection 3.2. just as recommended. It is summarized as much as possible. . You can check it in the final manuscript where your comment has already been taken into account.**

You can find my minor details as comment boxes in the pdf document.

**All comments included in the attached PDF have been taken into account. . You can check it in the final manuscript where your comment has already been taken into account.**

**Once again we want to thank you for your dedication and time to give us these comments. It is greatly appreciated**

---

## Author Comment (AC3)

RC3: 'Comment on hess-2024-82', Anonymous Referee #3, 28 May 2024

**Comments of the Referee#3.**

Ojeda et al. conducted interdisciplinary studies in Alhama de Aragón and Jaraba in Spain including the analysis of geologic and hydrogeologic conditions, the hydrochemical data, and groundwater modeling. Authors aimed to unravel the source of springs in the study area which would be helpful to the sustainable conservation strategy. Authors do use a lot of different kinds of data and build a likely sounding flow model. I think the results are important to understand the groundwater movement in the study area and this study is a good contribution. However, I don't think the current manuscript is well prepared for publication in HESS. The reasons are as follows:

1. The biggest problem is that the author didn't well leverage the model they built. If the objective is to identify the source of the springs, after you built the well calibrated flow model and did particle tracking using MODPATH based on your flow field, why not analyze the flow paths of the particle tracking results. Then you can easily get what you want and then the modeling work is essentially meaningful.

2. It looks like the authors' idea is that they propose a kind of conceptual model based on geologic and hydrogeologic conditions. Then they build a flow model using modflow. If the calibrated model has a good performance on different hydrologic variables by comparing with observations, then their proposed conceptual model sounds. I really cannot agree such an idea as you didn't correctly use the model and the modeling work lost its intrinsic significance.

3. If the author can utilize the model well, the last part of hydrochemisty is not necessary. You can merge them into your analyzation of your particle tracking results and use these data to validate your particle tracking results. In Lines 859 and 861, it is sad to see 'is assumed' as you still cannot identity the flow paths after si many modeling efforts.

4. So, the 'interdisciplinary studies' are like a documentation of all your work which do not connect each other tightly. The main line is not clear and a lot of descriptive sections just like filed work documentations.

**Reply to the comments of the Referee#3:**

Thank you very much for these observations that we consider important. The MODPATH was actually used at the time but we thought that in order not to accumulate an excessive number of figures in the article, it was finally not included. But we think it's very good to discuss this topic. So, we include the results in section 4.3.9, where it can be seen that the ages obtained from the flow considering the recharge area of the model are compatible with the results of the tritium isotopes for each spring, since they are of the same order of magnitude. We intended to write the following in the section of the article referring to the model (4.3.9.) and in 4.4. I hope they seem good to you:

We include the entire and updated chapter 4.3.9 where we have included all your comment:

In line 829 we will include the following paragraph:

4.3.9    Transit time estimation

Thanks to mathematical modelling of the hydrothermal aquifer, it has been possible to approximate the effective porosity of the hydrothermal system as well as the Darcy velocity.

In order to carry out this simulation, the Modpath package was used to simulate the path of a contaminant plume along the aquifer. This has allowed us to estimate the time it takes for a pollutant to travel through the aquifer until it emerges at the discharge points.

To achieve this, a contaminant located in the northernmost area of the calcareous outcrops of the Aragonese branch has been introduced, and another contaminant has been introduced in the outcrops of this same formation corresponding to the catchment area of the Deza springs.

We can see in figures 1-A and 1-B the progress of said tracer after 20 years and 80 years. It can be seen in Figure 1-A how after around 20-25 years it comes out through the Deza springs. And in Figure 1-B, how after 80 years. It leaves through the springs of Alhama de Aragón first, and then through Jaraba, having made a similar route concentrated in the deepest and most conductive area, although the peripheral flow that flows into Jaraba makes it slightly longer and older. Let us advance here the ages obtained by tritium from the following section, and although these have a semiquantitative and approximate value, we see that the results obtained by the two methods are of the same order of magnitude. Indeed, it is observed that the simulation shows that the waters of the Deza springs are more modern than those of Alhama and Jaraba, and that the age obtained from the model coincides with the time estimated by tritium from the latest analyses, which is about 20-25 years. In the case of Alhama and Jaraba we observe that the ages obtained by tritium from the most recent analyzes give us an age of more than 61 years, so 80 years could fit.

[Figure]

Figure 1-A                                    Figure 1-B

*Figure 1-A and 1-B. Simulation of the situation of a contaminant introduced in the recharge area of the outcrops of the thermal calcareous aquifer of the feeder (or catchment) basins of the Deza, Alhama de Aragón and Deza springs; Figure 1-A. Simulation of the contaminant plume situation after 20 years. Figure 1-B.  Simulation of the contaminant plume situation at 80 years of age.*

Finally, we see in Figure 2 the situation of a tracer stain along all the edges of the aquifer and also in the underground divide below the Tertiary after 250 years, with colors that indicate the age of the groundwater (where strong blue corresponds to the youngest ages and red corresponds to the oldest). Perhaps most prominently, in the SE zone of the aquifer, the recharge comes from the Tertiary, which has little influence on the magnitude of the flow and is very old water. On the NE side closest to the Aragonese Branch the flow is hydrodynamically more active and the water more modern.

[Figure]

*Figure 2. Groundwater age distribution*

Please keep in mind that these figures have been made quickly to respond to your comment quickly, the final ones will be better modeled

In addition, we have included in section 4.5. after line 1020, the following paragraph in red.

7. Nor are the values of groundwater age obtained from tritium for Alhama/Jaraba justified because 1015 the flow distances from the centre of gravity of the Sierra del Solorio to these springs (about 25 km) are not so large as to result in ages above 60 years, as occurs in our model with flow lines of 70 km. Neither does it explain the order of increasing age of Deza-San Roquillo-Embid- Alhama/Jaraba, it would have to be the opposite, as the length of the flow lines increases towards Deza. Furthermore, if the flow first passed through Jaraba and ended in Alhama (ITGE-DGA, 1994), the age of the latter 1020 spring would be greater, and yet its age and evolution is totally parallel to that of Jaraba. To validate these results, monitoring contaminant particles using the flow model obtains ages of the same order of magnitude as those obtained using the tritium data. In this way, the hydrochemical aspects acquire all their value when connected to the results of flow modeling.

**Comments of the Referee#3.**

Also, there are too many names which are not well introduced in the manuscript, and it is really messy and hard to follow the many descriptive words. For example, the most important 'Alhama de Aragón and Jaraba' even did not appear in your Figure 1.

There are also a lot of small errors everywhere in the manuscript and make it even harder to follow. For example, I don't think the numbers in the caption of Figure 5 are right.

**Reply to the comments of the Referee#3:**

You are right. These typographical errors have already been corrected, since they have been comments from Referee#1. A general and exhaustive review of the manuscript has been done. Names that did not appear before are added to figure 1.

**Comments of the Referee#3.**

> There are also a lot of small errors everywhere in the manuscript and make it even harder to follow. For example, I don't think the numbers in the caption of Figure 5 are right.

**Reply to the comments of the Referee#3:**

A general and exhaustive review of the manuscript has been done in order to avoid typographical errors. Figure 5 has been removed according Referee#2 comment.

**Comments of the Referee#3.**

Line 781: I don't think Figure 14 is the right figure you want to direct the audience to. Line 814: what is Fig. 16 and Fig. 16.

**Reply to the comments of the Referee#3:**

Due to the complex model that has been created and given the high number of boundary conditions, we believe that it is essential that Fig. 14 be included. In addition, this figure shows the location of the springs and other important boundary conditions which makes it easier for the reader to understand the model.

Line 814: what is Fig. 16 and Fig. 16: It is a typographic error. It should we said "Table 2 and Fig.15). In addition, a general review of the manuscript is made to check that there are no more typographical errors such as the one indicated.

**Comments of the Referee#3.**

Fig. 18: I don't think the legend is right. The red and blue points are neither observed values nor the simulated values.

**Reply to the comments of the Referee#3:**

The figure shows the maximum and minimum observed values, as shown in the table 2 and table 3. Please check it. I surrounded in a red square the minimum and maximum observed values concerning table 2.

790    **Table 2.** *Calculated vs observed groundwater heads at calibration points.*

| Alhama de Aragón piezometer | | Embid de Ariza piezometer | | Deza piezometer | |
|---|---|---|---|---|---|
| Calculated values | Observed values | Calculated values | Observed values | Calculated values | Observed values |
| 670 | 665 – 667.5 | 791 | 777-773 | 920 | 919 – 925 |

As can be seen, the calibration of the observation points is considered acceptable, as the measured and simulated values are very close (Table 2 and Fig. 17.).

In addition, we have changed the figures slightly so that they can be understood better. I show below the updated figures.

[Figure]

**Comments of the Referee#3.**

Line 827: what does "in order to carry out this simulation" mean? Does 'this simulation' mean your modflow model? The modflow model does not necessarily depend on your particle tracking.

**Reply to the comments of the Referee#3:**

As previously explained, a simulation was carried out with MODPATH, which we include in these answers

**Comments of the Referee#3.**

Line 946: you have the first half of the parenthesis, so where the second half? There are a lot of such small errors.

**Reply to the comments of the Referee#3:**

It's a typographical error. The manuscript has been reviewed and will be reviewed again to avoid these types of errors.